

Texture analysis of experimentally deformed Black Hills Quartzite
Rüdiger Kilian[1*], Renée Heilbronner[1,2]
(1) Department of Environmental Sciences, University Basel, Basel, Switzerland
(2) Department of Geology, University of Tromso, Tromso, Norway
* ruediger.kilian@unibas.ch
**Abstract**
The textures of three samples of Black Hills quartzite (BHQ) deformed experi-
mentally in the dislocation creep regime 1, 2 and 3 (according to Hirth and
Tullis, 1992) have been analysed by EBSD. All samples were deformed to relat-
ively high strain, within a temperature range of 65° and identical displacement
rates and are almost entirely composed of  dynamically recrystallized grains.
A texture transition from peripheral c-axes in regime 1 to a central c-axis max-
imum in regime 3 is observed. Separate pole figures are calculated for different
grain sizes, aspect ratios and long axis trend (θ) of grains, and high and low
levels of intragranular deformation intensity as measured by the grain kernel av-
erage misorientation (gKAM). Misorientation relations are analysed for different
texture components (named Y- B- R- and σ, with reference to previously pub-
lished 'prism', 'basal', 'rhomb' and 'σ1' – grains).
Results show that regime 1 and 3 correspond to clear end member textures with
regime 2 being transitional. Texture strength and the development of a central c-
axis maximum from a girdle distribution depends on deformation intensity at the
grain scale and on the contribution of dislocation creep which increases towards
regime 3. Combined with calculations of resolved shear stresses and misorienta-
tion analysis, it becomes clear that the peripheral c-axis maximum  in regime 1 is
not due to deformation by  basal - <a> slip. We interpret the texture transition as
a result of different texture forming processes, one being more efficient at high





stresses (formation of grains with peripheral c-axes), the other depending on
strain (dislocation glide involving prism and rhomb slip systems), and not  as a
result of a temperature dependent activity of different slip systems.
Keywords:
Quartz texture, crystallographic preferred orientation, texture transition, slip
systems, dislocation creep
**1. Introduction**
Quartz textures, usually presented in the form of pole figures are used frequently
for the analysis of deformed rocks. Interpretations based on pole figures or EBSD
data are widely used to make inferences about deformation kinematics such as
shear senses (e.g. Berthe et al., 1979, Simpson, 1980; Kilian et al., 2011b), vorti-
city (e.g. Wallis, 1995; Xypolias, 2009) and progressive strain type (e.g. Price,
1985; Sullivan & Beane, 2010), deformation mechanism (Behrmann & Mainprice,
1987; Song & Ree, 2007; Kilian et al., 2011a) or  recrystallization processes (e.g.
Knipe & Law, 1987; Stipp et al. ,2002), involved slip systems (e.g. Bouchez &
Pecher, 1981; Schmid & Casey, 1986; Law et al., 1990) or synkinematic temper-
ature (e.g. Kruhl, 1998; Morgan & Law, 2004; Thigpen et al., 2010). However, in
some cases, the underlying mechanisms and processes are poorly understood
and dependencies, e.g. of temperature and recrystallization mechanisms (e.g.
Stipp et al., 2002), or texture geometry and strain in polycrystalline materials are
not always easily separated (e.g. Schmid & Casey, 1986; Wenk & Christie, 1991).
Transitions in texture types have been correlated with (changing) recrystalliza-
tion mechanisms or were explained by a temperature dependence of the slip sys-
tems involved in crystal plastic deformation (e.g. Tullis et al., 1973). There have
been speculations on a temperature dependence of slip systems, either caused
by a temperature dependent critical resolved shear stress during glide (Hobbs,
1985) or anisotropic diffusion during climb (e.g. Blacic, 1975), however, conclus-





ive evidences have only been found for a transition from <a> to <c> burgers vectors towards very high temperatures (e.g. Mainprice et al., 1986). For <a>-slip, a temperature dependent activation of different slip systems has not been convincingly demonstrated. A  bulk strain dependency of texture is recognized in experiment and nature (e.g. Heilbronner & Tullis, 2006; Pennacchioni et al., 2010), however, it is not too clear in which way bulk strain relates to strain at a grain scale in a deforming aggregate.

In this contribution we will focus on the following questions: Which factors influence the texture geometry (shape of pole figure skeletons)? Is the texture controlled by deformation temperature, geometry/kinematics or recrystallization processes?  How reliably can certain texture components be used to infer the activity of a specific slip system? To this end, EBSD data obtained from Black Hills Quartzite (original samples of Heilbronner & Tullis 2002 and 2006) experimentally deformed in the three dislocation creep regimes (Hirth & Tullis, 1992) were examined. Regime 1 is characterized by a high yield strength and substantial strain softening and non-recrystallized grains deforming by fracturing and dislocation glide (Hirth & Tullis, 1992) and climb (Stipp & Kunze, 2008) while recrystallization occurs by bulging or nucleation and growth of new grains. Aggregates of newly formed grains are thought to deform by a dislocation process with a substantial contribution of grains boundary sliding (Tullis, 2002; Stipp & Kunze, 2008). Regime 2 samples yield at lower stresses and no pronounced weakening is observed. Incipient subgrain rotation recrystallization (SGR) has been documented (DellAngelo & Tullis, 1989). Regime 3 exhibits the lowest flow stresses, SGR is predominant, and some workers observe synkinematic normal grain growth (Gleason et al., 1993; Stipp et al, 2006) or abnormal grain growth (Heilbronner & Tullis, 2006), potentially in relation to texture development.



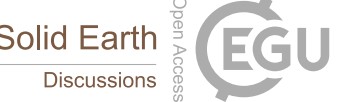

Specific types of textures and certain texture components have been given ge-
netic or descriptive terms in the literature. In particular,grains with a specific c-
axis direction have been interpreted to be suitably oriented for the activity of
specific slip systems with the <a>-direction as a Burgers vector and have there-
fore been called 'basal-', 'prism' or 'rhomb-grains' (Bouchez & Pecher, 1981;
Heilbronner & Tullis, 2006).  Here we will call grains with the c-axis at the peri-
phery of the pole figure, and approximately orthogonal to the shear plane B-
grains or B-domains, those contributing to the intermediate parts of a single
girdle, between the periphery and the centre will be termed R-grains or R-do-
mains, those with peripheral c-axes inclined against the sense of shear, roughly
towards the direction of the loading piston will be termed σ-grains or σ-domains
and finally  those grains with c-axes close to the centre of the pole figure Y-grains
or Y-domains on account of  their proximity to the Y-direction in the inferred
strain reference frame.
**2. Methods**
2.1 Experiments & samples:
The analysed samples are experimentally deformed Black Hills Quartzite (BHQ)
of Heilbronner & Tullis (2002 and 2006). 1 to 1.5 mm thick slabs of BHQ were
deformed in a solid medium, modified Griggs-type deformation apparatus. The
slabs were placed between two 45° pre-cut forcing blocks made up of single
crystal Brazil quartz oriented with the c-axis parallel to the advancing load pis-
ton. Experiments were performed on "as-is" BHQ and with 0.17 wt. % $H_2O$ ad-
ded (~11000 ppm $H/10^6 Si$) in mechanically sealed PT jackets. "As-is" experi-
ments were conducted at 850 °C and water added experiments at 875 and
915°C, all an axial shortening rate of ~$3*10^{-5}$ $s^{-1}$ and confining pressures of
1.5 GPa. Details of the experimental procedures are provided in Heilbronner &
Tullis (2002, 2006) and in the companion paper Heilbronner& Kilian (this
volume). Sample strain or flow related reference directions such as the principal

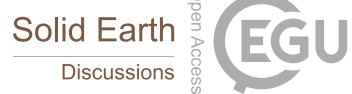

strain axes or the instantaneous stretching axes(ISA) were calculated from final
displacement, initial and final sample thickness assuming steady general shear
(e.g. Fossen & Tikoff, 1993).
2.2 EBSD data analysis
EBSD maps were collected on a Zeiss Merlin FEG-SEM, equipped with am Ox-
ford (insert. details here) EBSD Camera in low vacuum mode, using a 2x2/4x4
binning, 20 kV acceleration voltage, and a probe current of 6-9 nA using step
sizes of 0.5 and 1 µm and exceptionally 0.25 µm. Unless otherwise specified, only
the maps of the high bulk strain experiments w1092, w946 and w1092 were ana-
lyzed (details are given in Heilbronner & Kilian, this volume). Data cleanup and
all processing was done using the mtex toolbox by Ralf Hielscher (Bachmann et
al. 2011; Mainprice et al., 2014; https://mtex-toolbox.github.io/). See appendix
for details on data processing.
Textures are presented in the form of (inverse) pole figures using the point group
'321' for quartz (Fig. 1). Pole figures are displayed such that normal to the shear
zone boundary (surface of the forcing blocks of the experiment ) is vertical and
the displacement direction horizontal. Texture strength is given as texture index,
pole figure J-index and maxima of pole figure densities. See appendix for details
on texture calculations. Crystal directions or poles to planes of (0001),[0001],<
11-20>, {10-10}, {10-11} and {01-11} are abbreviated as (c), [c], <a>, {m},
{r}, and {z} with the conventional bracketing scheme.
Positions on the pole figure will be given as azimuthal angles with an origin in
the west, increasing clockwise, inclination being 0 in the centre of the pole fig-
ure. Directions in the first and third quadrant (NW and SE) are inclined with a
sinistral sense of shear.
Textures have been calculated for classes of grains in aspect ratio - grain size,
and long-axis trend (θ) - aspect ratio space. Grains from different maps of
identical step size of low and high total strain experiments of each regime have

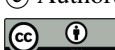



been combined to obtain a sufficiently large datasets. Class boundaries are chosen such that they are equally populated for each property and combinations of two properties result in a 3-by-4 matrix in e.g. aspect ratio – grain size space. Because distributions are skewed, classes are note equally populated. Kernel parameters for the texture calculations were estimated individually for each class and since that, pole figure geometries are comparable and densities of pole figures will not be overestimated for poorly populated classes.

Spherical interpolations for grain size, aspect ratio and axial ratio have been calculated in c-axis pole figure space. They represent the average grain property at a given c-axis direction. To avoid a bias introduced by the uneven distribution of c-axis poles the following procedure was used: A subsample of 400 grains was drawn, the c-axis direction was calculated, the property associated with each grain was interpolated on a 15° spherical grid using an inverse distance weighting, the procedure was repeated 1000 times and the mean of the interpolation was plotted as a pole figure.

Grain sizes are defined as the diameter of an area equivalent circle of the grains. The grain long axis and aspect ratio are obtained from the best fit ellipse from smoothed grain boundaries.

To quantitatively compare texture strength of the grain property classes (e.g. grains size – aspect ratio classes), subsamples of identical sizes were estimated in a bootstrapping approach. 100 randomly chosen subsamples of the size of the smallest of any population (>100) were repeatedly drawn from the population of grains within a property class. Texture parameters were calculated for each subsample using a fixed kernel width. The mode of the resulting distributions is compared. The standard deviation usually converged after <50 draws to below 5-10%. Texture strength was measured using the texture index (texture or J-index, L2-norm of the ODF) and two different fibre volumes. A fibre is defined by a crystal direction and a corresponding direction in specimen coordinates and manifests as a line in ODF space. The fibre volume is the mass fraction of an ODF contained within a radius around a given fibre. One fibre (B-fibre) is defined by the



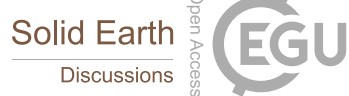

[0001] direction inclined 76° to the shear plane with the sense of shear (B-fibre)
and another fibre (Y-fibre) is defined by the [0001] direction pointing towards the
centre of the pole figure (structural y-direction). For easier inspection and com-
parison, results are colour-coded and plotted in x-y parameter space (aspect ratio
- grain size or θ - aspect ratio).
Misorientation axes have been determined in specimen and crystal coordinates
for misorientation angles of 2-9° for subsets of grains. Subsets a chosen to con-
tains grains with modal orientations of up to 25° away from the orientation
modes for Y-, B-, R-, and sigma-grains. This threshold angle corresponds to a
volume in the ODF and not to the opening angle of a cone on a [c] pole figure,
making the selection stricter compared to Heilbronner & Tullis (2006) since
thresholds are applied to modes of orientations and not only c-axis directions.
Pole figures and inverse pole figures are contoured as well as a random sub-
sample of points is plotted. Individual points are colour-coded such that misori-
entation axes in specimen reference frame appear in the colour key of the axis in
the corresponding crystal reference frame. Misorientation axes in crystal refer-
ence frame are colorised by the inclination of the axis in specimen coordinates.
<a>-intransparency is defined as the minimal angle between <a> across a grain
boundary, ignoring the polarity of <a> and it is a measure of strain compatibility
of adjacent grains deforming by glide along <a> on an infinite group of planes.
The grain kernel average misorientation (gKAM) is calculated from the kernel av-
erage misorientation (KAM) from noise reduced EBSD data. The gKAM is the
sum of the KAM within a grain divided by the number of measurements with the
grain. The gKAM is a measure of intragranular deformation intensity or misori-
entation density and depends on the misorientation angle and fraction of low
angle boundaries within a grain. See appendix for details.



The generalized Schmid factor Sf (Reid, 1973) is calculated from a given slip sys-
tem and a stress tensor, and presents the ratio between the shear stress on a slip
system and the norm of the macroscopic stress tensor. Since the general shear
experiments are plane strain and displacement is resolved parallel to the dip of
the forcing block, we use a triaxial, normalized stress tensor. The absolute mag-
nitude of the stress tensor does not have any influence on the value of the gener-
alized Schmid factor. Schmid factors are calculated for all orientations (either
each measurement or grain modal orientation), and their sum is divided by the
number of orientations. For combinations of slip systems, Schmid factors are cal-
culated for all slip systems in the combination and the maximum values are aver-
aged.
**3. Results**
3.1 Pole figure geometry
In the regime 1, the pole figures show a broad, asymmetric peripheral distribu-
tion of c-axes with a maximum ~78° (inclined with the sense of shear) (Fig.1a).
Minor densities occur at ~130° roughly parallel to the shortening axis in the ex-
periment and a tail towards the centre of the pole figure.  <a> shows a major
maximum at the periphery, forming an angle of ~-12° with the shear plane. Two
minor maxima of <a> lie on great circles inclined about 15° with respect to to
the pole figure centre. Poles to {r} shows a symmetric peripheral maximum in-
clined against the sense of shear at ~+120° and a girdle distribution perpendicu-
lar to the peripheral maximum.
In the regime 3 sample, c-axes pole figures show an elongated maximum in the
centre of the pole figure, overlying a weak, kinked single girdle. Internally the
maximum is composed of two maxima at an angular distance of about 20°, sym-
metrically arranged above and below the shear plane. A-axes form a major max-
imum at ~-10°.



In pole figures obtained from regime 2 experiments, [c] is distributed along a
kinked single girdle, presenting a combination of regime 1 and 3 pole figures. A-
axes form a strong maximum at ~-13° and the {r} pole figure resembles the one
observed in the regime 1 sample.
Minor but significant densities of [c] in the pole figure are positions inclined
against the sense of shear (roughly corresponding σ-directions) are only found in
the regime 1 samples.
Inverse pole figure (IPF) have been constructed for various strain and sample re-
lated reference directions (Fig. 1b): in general and most dominant in regime 1
and 2, the strongest alignment is found for <a> and a reference direction of the
ISA1 - 45°, ( -10° below the shear plane, being the trace of the highest shear
stress). In the regime 3 sample, <a> also shows a strong alinement with the
shear direction since it is generally more strongly dispersed around the peri-
phery. In IPFs with a reference direction at 135° (parallel to the direction of the
load piston), all samples show a high density very close to {20-21}. Using the
shortening ISA (being steeper than the direction related to the load piston) as a
reference direction, in regime 1 and 2 a very strong alignment of {r} is found.
3.2. Orientation maps.
Based on the result of the IPFs, we display orientation maps in an inverse pole
figure colour coding with respect to the ISA1-45° (Fig. 2a).  Maps show a relat-
ively homogeneous distribution of <a> parallel to the reference direction across
all three samples. Notably in regime 1 and 2 samples, inside bands with a shear
band geometry,dispersed grains show an alignment of [c] close to the shear
plane.



Using the structural Y-direction as a reference (Fig2b), the increase from regime
1 to 3 in orientations with [c] parallel Y can be clearly seen. In regime 2 it ap-
pears that tY-grains dominate in areas mostly devoid of larger porphyroclasts and
a smaller grain size. In regime 3 Y-grains also occur in a spatially domainal struc-
ture.
3.3 Variation of pole figure geometry within classes of grain size, aspect ratio
and long axis direction
For pole figures calculated in aspect-ratio-grain size space, the following obser-
vations are made (Fig.3): For all regimes, [c] pole figures show and increased or-
dering towards higher aspect ratio and towards larger grain sizes. In regime 1
small grains in general show the broadest distribution, dispersed along the peri-
phery.  With increasing aspect ratio, a weak single girdle can be recognised. In
regime 3, [c] pole figures obtained from large grains with the large aspect ratios
show the highest degree of ordering with the elongated central maximum and
peripheral maxima and minor off-periphery maxima, forming a single, kinked
girdle. In pole figures calculated for regime 2 experiments, the identical relation-
ship is observed, a single kinked girdle develops with peripheral and central
maxima, which can be described as mixtures between regime1 and regime 2 pole
figures (Fig.3).
For pole figures of θ – aspect ratio classes, in general those [c] pole figures from
within classes of the largest aspect ratio and θ containing the maximum of the
trend distribution have the highest degree of ordering.  Second strongest [c]
alignments are found for regime 1 in the θ class which contains the trend of the
shear plane and for regime 2 and 3 the θ class with steeper major axis trends.
The weakest [c] ordering can be found in those classes containing grains with a θ
pointing against the sense of shear. While the distribution of maxima along the
girdle characteristically varies from regime 1 over regime 2 to 3, additionally

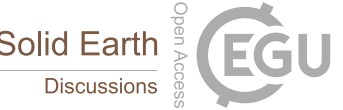

within each regime, the classes with the highest aspect ratios and within those,
the θ class for most well aligned grains, show an increasing concentration of [c]
in central regions of the girdle.
The position of the peripheral part of the kinked girdles shows a consistent vari-
ation with the θ class, being most inclined (with the sense of shear) in those
classes also containing the trend of the shear plane and being steepest within
the classes of θ steeper than the maximum (table 1). This variation is most ex-
pressed for classes of high aspect ratio.
To summarize, the pole figure skeleton, the density along the girdle and its in-
clination varies with θ and the deviation of [c] maxima on the girdle from the
periphery depends most on grain aspect ratio.
Pole figures for <a> and {r} for the different classes of grain size, aspect ratio
and θ are far less subject to changes in geometry and exemplified for regime 2
samples (Fig. 4). <a> readily form peripheral maxima close to the shear plane
and ordering increases with increasing grain size and aspect ratio. Pole figures
for {r} show the peripheral maximum at ~110-120° which varies together with
the trend of the [c] girdle (Table 1).
In a few cases, a secondary, peripheral [c] maximum (or relict cross girdle) is
present and its opening angle varies between 50° to about 80°, mostly as a func-
tion of aspect ratio but not systematically between regimes.
3.4 Pole figures of grain properties
For pole figures of grain size, aspect ratio and axial ratio (Fig. 5) obtained from a
subset of grains smaller 25 µm, for regime 1 and 2 the average grain size (com-
parable to a number weighted average grains size) is largest at the periphery
and for regime 3, it is largest in the centre of the [c] space. In all three regimes,



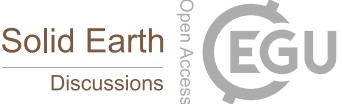

a high average aspect ratio is found along a kinked girdle for regime 1 and 2 and
a weak cross girdle for regime 3. The largest average aspect ratios along these
girdles are located at the centre of the girdle for regime 2 and 3. Inversely, the
largest average axial ratio occurs for grains with [c] in one of two close to ortho-
gonal, peripheral directions, roughly at 30-35° or 125°.
3.5 Pole figures for low-gam/high gam grains
Pole figures are calculated for populations of grains with a gKam below and
above the median gKam, independent on grain shape or size parameters. The
most obvious difference between the textures for low and high gKam classes are
seen in the c-axis pole figures for recrystallized grains (Fig. 6a). Here, grains <
12 µm for regime 1 and 2 and < 25 µm for regime 3 are considered. In general
[c] pole figures for the high gKam population show a stronger degree of ordering
and a tendency for higher pole densies away from the periphery, along the
kinked single girdle. The higher degree of ordering is expressed by an creasing
the pfJ, higher densities along the girdle, narrower peripheral maxima and in the
case of of single maximum pole figures also by the magnitude of the maximum.
Peripheral maxima shift to Y-maxima (regime 2) or secondary peripheral maxima
disappear in the high gKam classes. This trend is more pronounced in textures
calculated from all the orientations from within grains within a given class. For
<a> and {r} pole figures, mostly a strengthening of the maxima at the periphery
can be noticed. Comparing pole figures obtained from the uppermost and lower-
most 20% of the gKam population, this change in geometry is even more pro-
nounced (Fig.6b).
3.6 Quantification of texture strength
In all regimes, the texture index increases with increasing aspect ratio and for
higher aspect ratios also for increasing grain size (Fig. 7). Also, the classes con-
taining the (recrystallized) grains with the largest size and aspect ratio possess
the highest texture strength. In each regime and class, there is also an increase
in the texture index from those population of grains with a low gKAM to those





with a high gKAM (0.05-0.55 and >0.55°). For regime 2, there is an additional
maximum for small grains with high aspects ratios within the higher gKam class.
Within θ - aspect ratio space, it is observed that the texture index continuously
increases towards higher aspect ratio. In regime 1 the highest texture index is
found for high gKAM classes at θ of ~20-30°. In regime 2 the range of the tex-
ture index is smaller but still maximum values are found for high aspect ratio
classes with a θ containing the shear plane or in the class of 20 to 40°. For re-
gime 3, also high aspect ratio classes have the highest texture index with the
maxima found clearly off the shear plane in the 30-45° θ bin. For all regimes,
there is an increase in the maximum texture index form the low gKAM to the
high gKAM class.
For aspect ratio - grain size classes and θ - aspect ratio classes the volume of the
B-fibre is largest for regime 1 and smallest in regime 3 while in contrast, the
volume of the Y-fibre is smallest in the regime 1 samples and largest in regime 2
samples, reflecting what can be roughly seen in the pole figures. The variation of
B-fibre volumes within regime 1 and y-fibre volumes within regime 3 is compar-
able to trend in variation of the texture index in the corresponding classes; also
with an increase in fibre volumes form low gKAM to high gKAM classes in θ - as-
pect ratio space for regime 1 and regime 3 as well as in aspect-ratio - grain size
space in regime 3. Regime 2 does not show a large variation in B- and Y-fibre
volumes (since c-axis girdles host both components), however, there is a small
decrease in volumes of B-fibres and in increase in the volumes Y-fibres from low
gKAM to high gKAM classes.
3.7 Structure of Y-domains and misorientation axes related to low angle boundar-
ies
Figure 8a shows a crop of an EBSD map of a Y-domain with a colour coding of
boundaries based on the <a>-intransparency. While c-axes are strongly aligned,
the <a>-intransparency can have relatively high angles (>20°) and may change



gradually along grain boundaries. Low values of the <a>-intransparency
between grains with c-axes at a high angle to another are also present, however,
less frequent than low values of the <a>-intransparency between y-grains. Grain
boundaries of the Y-grains show in total a larger deviation from the uniform dis-
tribution of the <a>-intransparency with more lower and more higher angles for
boundaries between Y-grains (see Appendix A2), while inter the latter approach-
ing a distribution which would be expected for a uniform texture.
colour-coding low angle boundaries for their misorientation axis in crystal co-
ordinates shows that most low angle boundaries within a y-domain have a rota-
tion axis close to the c-axis (Fig. 8b). However, other directions are also present
and in non-Y grains, rotation axes close to one of the poles to rhombs or of direc-
tions located within the basal plane appear to be more frequent.
Misorientation axes are shown for a regime 2 sample for Y-, B-, and R-grains
(Fig. 9). For the complete dataset for all three regimes including sigma grains
see Appendix 4. Misorientation axes dominate around the c-axis direction, also
coinciding with the axes being most inclined (parallel to the kinematic y-direc-
tion). For B-grains, the highest density of misorientation axes is found to have
directions within the basal plane, mostly close to m. However, the strength of
this distribution is very weak. R-grains show a maximum of misorientation axes
around c and a slightly higher density in the area of positive rhombs. Misorienta-
tion axes with direction most closely parallel to the kinematic Y-direction prefer-
entially fall close to a position between the <10-11> and a more general direc-
tion ~ <7-2-56>. For σ-grains, a distribution is found similar to that of the b-
grains, however with a slightly more pronounced deviation from uniformity (see
Appendix 4). For all grain classes it can be seen that a variably strong maximum
of misorientation axes in specimen coordinates is parallel to the kinematic Y-dir-
ection. R-grains show the highest density of misorientation axes in crystal co-
ordinates around the c-direction while in specimen coordinates, the maximum is
located also at the kinematic Y-direction. Since this is in the first place a contra-

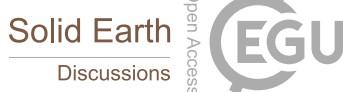



dictory situation, is needs to be noticed that the distribution of misorientation
axes in specimen coordinates is elongated towards axes which correspond to ro-
tations around the c-axis (red clusters, third column in Fig. 9d), while those mis-
orientation axes located at the centre of the pole figure correspond those crystal
directions loosely located between 10-12, 10-11 and <7-2-56>.
3.8 Schmid factor analysis
The mean generalized Schmid factor is plotted as function of the trend of the
maximum principal stress direction of the stress tensor (Fig. 10). Highest Schmid
factors are attained for σ1 directions consistently about 10 to 20° steeper com-
pared to the direction of the load piston. For single slip systems, in all regimes,
disrespect of whether modal grain orientations or all points of the EBSD maps
are used, {pi'}-<a> or {z}-<a> give the highest Schmid factors. Notably also in
regime 1 where many grains have [c] at the periphery of the pole figure, highest
mean Schmid factors are predicted for {pi'}-<a>. For combinations of slip sys-
tems, in regime 2 and 3, {m}-<a> + {pi'}-<a> + {z}-<a> always give the
highest mean Schmid factors and are equally high in regime 1 as the combina-
tion of {m}-<a> and (c)-<a>. Notably, the curve for {m}-<a> shows a similar
behaviour as the curve for <a> slip on the positive rhombs, having a minimum
mean Schmid factor at the position where for most reasonable slip systems show
a maximum. While this is logical for {r} and {z} for example, {m} have this crys-
tallographic dependency and this behaviour is somewhat unexpected.





### 4. Discussion

Textures of recrystallized grains and bulk textures of all regimes share an alignment of <a> to the direction of parallel to the trend of the plane of maximum shear stress at ~-10° within the shear plane and a strong alignment of the positive rhombohedral planes towards the shortening ISA (regime 1,2). There is a transition in texture geometry (skeleton) and in texture strength across the regimes. [c] is dispersed on the periphery normal at a high angle to the shear plane in regime 1, distributed along a kinked, single girdle in regimes 2 an forms a central bi-modal maximum at the structural Y-direction in regime 3. The texture strength increases from regime 1 to regime 3.

Similar pole figure skeletons have been reported in nature (e.g. Bouchez & Pecher, 1981; Mancktelow, 1987; Law et al., 1990) and similar texture transition were observed within metamorphic gradients (Stipp et al., 2002). Occasionally this type of transition was used to draw inferences on the metamorphic conditions for mylonitisation (see Law, 2014 for a review). In experiments, a texture transitions was observed in axial compression experiments (Tullis et al., 1973), where in the high stress regime [c] pole figures have a single maximum parallel to the compression axis and in low stress regime, [c] occurs within a small circles centred around the compression axis. Some types of texture transitions observed in nature (e.g. Lister & Dornsiepen, 1982; Gapais & Barbarin, 1986) were often explained to be related to the activity of prism-c slip at very high temperature (e.g. Mainprice et al., 1986). Based on the speculation on a temperature dependency of different <a>-slip systems (Blacic, 1975; Hobbs, 1985), certain types of textures were thought to be related to <a>-slip on different planes. However, those texture transitions with evolving densities along a single or cross girdle (e.g. Stipp et al., 2002, Toy et al., 2008) are difficult to explain with a hypothesis of temperature sensitive <a>- slip systems and factors such as strain, alternative texture forming processes and the influence of recrystallization mechanisms are variables that need to be taken into account as well.

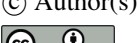



In the analysed experiments, the temperature difference was only about 65° and
displacement rates and finite strains are roughly identical in all regimes and the
major difference observed for these samples is the peak and flow stress (Fig. 2).
Accordingly, a temperature and bulk strain rate dependence can be neglected,
and the effects of the recrystallization and texture forming mechanisms of
samples deforming at basically identical conditions but different flow stresses
can be studied.
4.1 Relation of the texture transition to deformation intensity at the grain scale
Increasing central densities along a (partial) girdle are documented especially
for regime 2 and 3 with increasing grain scale deformation. We will take the θ –
aspect ratio relation and the gKAM as a measure for grain scale deformation.
Based on the coincidence of the maximum in the distribution of θ with the ISA
and the observation of the most synthetically rotated girdles in the θ classes
which contains the direction of the long axis of the finite strain ellipsoid (Fig 3),
grain alignment is assumed to be related to deformation. Similarly, we assume
that grain lengthening is related to deformation. If strain at the grain scale is
achieved via a dislocation mechanism, and if low angle boundaries are the result
of dynamic recovery by a climb of the strain producing slip system,  the gKAM is
an expression of the intragranular deformation intensity. These assumptions are
supported by the observation of an increased texture strength with increasing
grain lengthening, alignment and increasing gKAM. We will argue that the re-
gime 3 sample shows the strongest support for dislocation creep and that dy-
namic recrystallization is dominated by subgrain rotation, and therefore a high
texture strength is associated with dislocation creep and this includes glide
which is the texture forming process.
With respect to the [c] pole figure skeleton an increasing ordering, development
or strengthening of a girdle component and/or the formation of a central max-
imum can be related to grain scale deformation. All these observations (Fig. 3,6)
indicate that [c] moves away from the periphery of the pole figure as grains de-



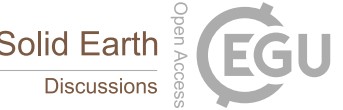

form. Given that the BHQ starting material shows a close to uniform texture, the
the Y-maximum is strain induced with [c] rotating along the single girdle towards
the centre of the pole figure. In nature, very similar textures evolutions are re-
ported as a function of bulk shear strain (Pennacchioni et al., 2010) and also
there, even at the highest observed shear strains, [c] only approaches the Y-dir-
ection. It remains to be explored whether this rotation is continuous or if there
are temporarily stable orientations. Positions of slow rotation rate might yield
the occasionally reported "rhomb"-maxima, although the exact c-axis positions
are quite variable and may be depend on other factors such as the kinematics of
flow as well.
In contrast, in regime 1 we observe a large volume of grains with [c] at the peri-
phery of the pole figure which must originated from the uniformly textured BHQ
and their occurrence cannot be explained by the observed strain related evolu-
tion of orientations.
## 4.2 Textures as indicators of deformation mechanism
### 4.2.1 Deformation mechanisms
Quantitative analysis of texture strength may give direct information on the con-
tribution of texture forming processes. Because an estimation of texture strength
depends on many variables, estimators should always be conservative in the
sense of not overestimating texture strength and because in geologic materials,
there is no knowledge about the exact meaning of absolute values, we performed
a quantitative comparison. Absolute densities and hence texture strength is ex-
pected to underrepresent the density of the underlying "true" distribution but re-
lative differences are quantitatively comparable in the presented approach (Fig.

484  7).

An increase in texture strength from regime 1 to regime 3 is documented as well
as a strengthening of the texture with increasing grain scale scale deformation.
The strength of a texture originating from deformation is usually related to the



contribution of crystal plastic mechanism such as dislocation glide and climb. At
the other hand, it can also be shown that crystal orientations and grain shapes
show a specific relationship since e.g. [c] girdles rotate as a function of θ (Fig. 3,
Table 1), which suggests that a certain amount of grain rotation takes place in
the forma of rigid body rotation and not entirely related to the internal deforma-
tion of the grains. Such a relative grain movement requires grain boundary slid-
ing. The necessity of grain boundary sliding during dislocation creep is well
known in some metals (e.g. Kottada & Chokshi, 2007) and has been suggested
for quartz mylonites (e.g. Mancktelow, 1987; Kilian 2011a) as a process related
to strain compatibility (Zhang et al., 1996). Glide with the involvement of a
prism-a slip system induces grain rotation around [c] within a Y-domain which
will result in strain incompatibility with the neighbouring grain as seen in the oc-
currence of large angles for the <a>-intransparency (Fig. 8a) which fits the sug-
gestion in the literature (Mancktelow, 1987) that this type of behaviour would be
expected for a texture with a Y-maximum.
In regime 1small grains have [c] broadly dispersed along the periphery (Fig. 6,7)
and with increasing aspect ratio and grains size of recrystallized grains, [c] gath-
ers towards the peripheral edge of a partial single girdle and the texture
strength is much lower compared to the regime 3 sample it is likely that the con-
tribution of grain boundary sliding in regime 1 is larger. Grain boundary sliding
has been suggested to significantly contribute to deformation in regime 1(Tullis,
2002; Stipp & Kunze, 2008). Possibly, in regime 1, the newly formed grains are
rather undeformed and may be smaller than the equilibrium subgrain size, lead-
ing to grain boundary sliding which correlates with the observation of the very
broad dispersion of [c] at the periphery for low gKAM grains in regime 1 (Fig.
6b).
In summary, the contribution of dislocation creep is interpreted to be largest in
regime 3 and the contributions of grain boundary sliding is largest in regime 1
and smallest in regime 3 samples. Since all mechanisms operate concurrently,
observed changes in texture strength are a result of different contribution of



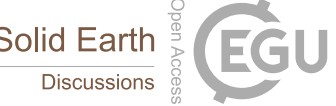

each individual process. Regime 2 is again envisaged to n an intermediate situ-
ation – more dislocation creep contribution than in regime 1 and more grains
boundary sliding contribution than in regime 3.
4.2.2 Recrystallization processes
Porphyroclasts usually show systematic substructures characterized by discrete
orientation domains of a size comparable to the recrystallized grains size. In re-
gime 2 and especially regime 3, orientation domains are recognized which are
roughly of the size of original BHQ grains (Heilbronner & Kilian, this volume) but
are fully recrystallized. The progressive change in grain orientation with respect
to their neighbours in Y-domains in regime 3 is compatible with rotation of parts
of the crystal around misorientation axes parallel to the vorticity axis inferred for
the experiment (Fig. 8 a,b). All these microstructures are compatible with SGR
recrystallization. In regime 1 where bulging recrystallization is dominant (Hirth
& Tullis, 1992, Stipp & Kunze, 2008) a large fraction of recrystallized grains at-
tain a new orientation, unrelated to a host derived orientation domain. Besides,
that the lack of an orientation relation with the host may also result from a con-
tribution of grain boundary sliding, smallest, most equiaxed grains have the
highest c-axis densities at a peripheral position (Fig. 5) and poles to {r} align to-
wards the shortening direction.  The exact nature of the process that controls the
orientations of newly formed grains during bulging recrystallization remains un-
clear and there are controversial suggestions in the literature (e.g. Stipp &
Kunze 2008; Cahn & Mishin, 2009).  For all larger grains, it is reasonable to as-
sume that the clustering of poles to {r} to {pi} (the latter closest to the elastic-
ally softest direction) at the shortening direction is the result of Dauphiné twin-
ning (e.g. Tullis & Tullis, 1972) caused by a rotation of {z} away from the short-
ening direction around the c-axis. Because Dauphiné twinning does not affect the
direction [c] and a fibre-texture around {pi} or {r} is not strongly developed, a

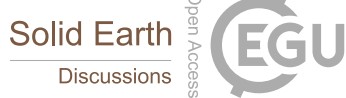
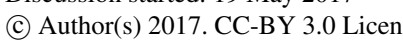


different process needs to be responsible for the preferred occurrence of [c] at
the periphery in regime 1. Preferred growth is one such process and the poten-
tial will be discussed within section 2.4.
4.2.3 Evidence of active slip systems
Although, the interpretation of slip systems based on misorientation carries un-
certainties because the direction of subgrain boundaries is not known from 2D
sections, it supplies more information compared to a purely pole figure or de-
formation lamellae based speculation. Most misorientations in y-domains (Fig 9.,
Appendix 4) are compatible with a dominance of a {m}<a> slip system and be-
cause axes in specimen coordinates coincide with the inferred vorticity axis,
there is a high probability of a tilt character for boundaries trending parallel to
{a}.
The R-domains show a distribution of axes in specimen coordinates which ap-
pears elongated towards the positions of the [c] in the pole figure. The superposi-
tion of the two populations of misorientation axes roughly parallel to c-axes res-
ults in a maximum density around [c] in crystal coordinates. This situation points
to the necessity to interpret density contoured misorientations axis plots care-
fully, because the representation in crystal coordinates honours crystal symmetry
which may result in high densities of superposed misorientation axes which are
physically different in specimen coordinates. Linking misorientation axes in both
coordinate systems via a common colour coding such as the one used here can
an example (Fig. 9) to overcome this problem. Misorientation axes in the R-do-
main which appear parallel to the vorticity axis reside in the position close to
<10-11>, <10-12> and ~<7-2-56>. The first and second direction are compat-
ible with {pi'}-<a> and {z}-<a> glide, respectively; both slip systems have been
identified in nature (e.g. Morales et al., 2011), experiment (e.g. Linker et. al.
1984) or suggested based on texture and Schmid factor considerations (Law et
al., 1990). Proximity to <7-2-56> could theoretically, in the case of a tilt charac-
ter, correspond to {s}-<c-a>which might be doubted to operate since the fairly



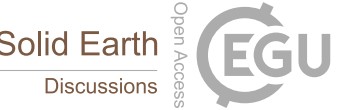

high length of the burgers vector. In general, misorientation axes of the R-do-
main are compatible with the oblique activation of {m}-<a> as well as any of the
{z}- or {pi'}-<a> slip systems. This interpretation is compatible with the per-
sistence of <a> aligned with the trend of the plane of highest shear stress while
[c] rotates towards the centre of the pole figure along the kinked girdle.
The misorientation axes of the B-domain show a distribution close to uniform. In
the case of (c)-<a> activity and the presence of tilt boundaries, misorientation
axes are expected to be clustered around <m> and in the case of conjugate slip,
to disperse in specimen coordinates along the trace of the basal plane. In the
case of twist boundaries, the rotation axes would be expected normal to (c),
which will be in the situation of the B-domains at a high angle to the shear plane.
Since neither of this is strongly pronounced, it might be that (c)-<a> is not one
of the slip systems being most active in the B-domain.
The above interpretation seems to be compliant with the Schmid factor analysis
(Fig. 10), where overall high Schmid factors are found for {pi'}-<a>, {z}-<a>
and partially {m}-<a> with an orientation of the maximum principal stress axes
aligned with the shortening ISA.
4.2.4 A model for the texture development and texture transition
As outlined above, a deformation-dependent rotation of [c] away from the peri-
phery of the pole figure is observed, it is found that most likely {m},{z} and
{pi'}<a> slip systems operate together and that evidences of (c)<a> are scarce
but there is no indication that any of the above described processes and mechan-
isms contributes to the presence of grains with [c] at the periphery of the pole
figure as mostly encountered in the regime 1 sample.
The principal difference in that suite of experiments is the sample strength. Be-
cause displacement rate constant and temperature differences are vey small in
the current experiments, it is  controlled  through the amount of water added to
experimentally deformed quartzites (e.g. Kronenberg & Tullis, 1984; Jaoul et al.,





1984). Despite early speculations that different water contents having an influ-
ence on the activity of specific slip systems (e.g. on synthetic single crystals,
Blacic, 1975), the role of water is mainly related to control the mobility of grain
boundaries in polycrystalline quartz aggregates (e.g. Gleason et al., 1993; Stipp
et al. 2006) as well well as in naturally deformed rocks (e.g. Mancktelow & Pen-
nacchioni, 2004; Kilian et al, 2016). The amount of water present in the the "as-
is" samples is still sufficiently high to hydrolyse dislocations even at high densit-
ies (e.g. Paterson, 1989) and the overall weakening effect will rather be a result
of the enhanced recovery processes. Accordingly, a direct relation of water to the
texture transition through a crystal plastic mechanism can be most likely ex-
cluded.
We suggest a model in which the texture transition can be explained based on
the hypotheses that a) during SGR and dislocation glide, [c] moves towards the
centre of the pole figure along seemingly well defined paths which constitute the
girdle and that b) an additional process operates that produces new grains (new
grain formation NGF) with [c] at the periphery of the pole figure at a high angle
to the shear plane which operates at higher stress levels. Both texture forming
processes compete, with the first one being dominant in regime 3 and the second
one being dominant at high stresses in regime 1. Regime 2 is a transition where
both contributions might be roughly balanced.
In regime 3, most SGR occurs and recrystallization is fastest, hence the rotation
of [c] towards the Y-direction will be happening most rapidly, NGF does not hap-
pen substantially, hence not many new grains are formed with [c] at the peri-
phery. All large grains are recrystallized in the high strain experiments. In re-
gime 1, NGF occurs most and SGR is slowest, the matrix of small grains deforms
with a fair distribution of grain boundary sliding, hence the rotation of [c] to-
wards the centre is slow and [c] is dispersed along the periphery. Because strain
partitions into the matrix, porphyroclasts that stretch most effectively are those
that survive longest.



Several lines of evidence support a mechanism such as NGF. There is ample in-
dication that growth of new grains after fracturation, or in most highly strained
crystals and in general under non-hydrostatic stresses during deformation can
result in a moderate to strong crystal preferred orientation in experiments (e.g.
Hobbs, 1968; Gleason et al., 1993; Vernooij et al, 2006; Trepmann et al., 2007;
Trepmann & Stöckert 2013) and in nature (e.g. Hippert, 1994; Hippert & Egydio-
Silva,1996; Menegon et al. 2008; Spiess et al., 2012; Kjöll et al. 2015). Newly
formed grains are found to have [c] roughly parallel and less commonly ortho-
gonal to the (inferred) shortening direction (e.g. Hobbs, 1968; Gleason et al,
1993; Trepmann & Stöckert, 2013; Kjöll et al. 2015), or roughly at 45° to the
stretching direction of shear fractures (e.g. Vernooij et al., 2006; Trepmann et la.
2007; Menegon et al., 2008). Fracture and microfracture development have doc-
umented in the Griggs-rig, even at high confining pressures and is thought to be
an essential process to enable crystal plasticity in the experiments (FitzGerald et
al., 1991; denBrok & Spiers, 1991; Stünitz et al., 2017). This might most likely
applies also to BHQ during initial steps of deformation and large porphyroclasts,
as well as quartzites deforming in the hardening regime with limited grain
boundary mobility (Hirth & Tullis, 1992). The positions of grains with high axial
ratios provides [c] directions (Fig. 5) which can be compatible with newly formed
grains, which have orientations subsequently modified by grain boundary sliding
and crystal plastic processes while they grow. The occurrence of shear band -
like features with c-axes roughly aligned with the opening/stretching direction of
the bands (Fig. 2) can be taken as indication that also a local kinematic frame-
work oriented grain grow with a preferred orientation. We do not have any direct
evidence of a fracture/nucleation/growth origin of grains newly forming in the re-
gime 1, however, in most experimental studies B-domains seem to be the first to
form at high stress conditions and e.g. the texture transition observed by Tullis
et al. (1973) perfectly matches our model with preferred growth of [c] parallel
and to a very minor amount perpendicular to the shortening direction. Whether
the resulting texture in regime 1 relates to anisotropic growth controlled purely





by nucleation and growth under non-hydrostatic stresses, influenced by the
elastic anisotropy of quartz or is additionally influenced by the local kinematics
of quartz must remain to be evaluated in future studies.
4.3 Peripheral c-axes are not due to (c)<a> slip?
Textures with peripheral [c] are often observed in rocks of low grade conditions
and based on the postulated temperature dependence of <a> slip systems, it was
assumed that (c)<a> may operate readily at low temperature conditions. The ex-
pectation that (c)<a> is an easy slip system in quartz relies on studies which ac-
tually did not demonstrate the existence of (c)<a> as an easy slip system. The in-
terpretations of the activity of (c)<a>, and hence the contribution to strain are
based on the presence of deformation lamellae as well as macroscopic sample
features, produced in experiments conducted in most pasts at extremely high
differential stresses (e.g. Christie et al. ,1964; Heard & Carter, 1968; Baeta &
Ashbee, 1969; Christie & Ardell, 1974; AveLallement & Carter, 1971). Despite
the assumption that deformation lamellae represent slip planes has been revised
several times (e.g. McLaren & Hobbs, 1972; White, 1973), they were still fre-
quently referred to as indicative of a specific slip system. There is ample evid-
ence for <a>-slip on {m}, {z} and {pi'} while (c)<a>glide evidences were miss-
ing (Christie & Green, 1964; Morrison-Smith et al., 1976; Twiss, 1976; Gapais &
White; 1982; McLaren, 1983; Linker et al., 1984) and many studies indicate that
(c)<a>-related dislocation systems are not found or are likely of very limited mo-
bility unless deforming by climb (e.g. Trepied & Doukhan, 1978; Doukhan &
Trepied, 1979; Trepied et al., 1980; Mainprice et al., 1986; Mainprice & Jaoul,
2009; Morales et al., 2011). It is beyond the scope of this contribution to dissect
the literature on quartz slip systems, but given that TEM based studies usually
find alternatives to easy (c)<a> glide, we will assume that neither purely pole
figure based based studies (Bouchez & Pecher, 1981; Schmid & Casey, 1986,
Heilbronner & Tullis, 2002; Kilian et al., 2011) nor models (Lister, 1979) can,
despite the intriguing geometry, be necessarily taken as evidence for large

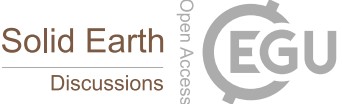

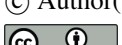

plastic strains accommodated by (c)-<a> glide. In the case of the origin of B-
grains by nucleation and growth or growth of specific fragments produced dur-
ing microcracking in a high driving force environment (high stress, high disloca-
tion density), the presence of B-domains are not related to the activity of (c)<a>.
4.4 Implications for interpretation based on quartz c-axis data
Often, the primary control in experiments to produce the different dislocation
creep regimes was temperature (Hirth & Tullis, 1992) and strain rate, although
the strength dependence of the different regimes can also be revealed in experi-
ments with different amount of water (Hirth & Tullis, 1992; Stipp et al., 2006). In
nature, microstructural transitions have been documented (Stipp et al., 2002)
whereas different recrystallization processes developed over a metamorphic
gradient and accordingly a correlation of temperature, respectively recrystalliza-
tion processes with a specific type of texture was tempting. There are wide-
spread examples of Y-maxima textures in natural quartz mylonites where the re-
crystallization mechanism is mainly subgrain rotation recrystallization with only
minor involvement of GBM (Mancktelow, 1987; Fitz Gerald et al., 2006; Pennac-
chioni et al., 2010) while in the observation of Stipp et al (2002) Y-domains star-
ted to appear in only in the GBM regime. Given our observation of the intensific-
ation of the Y-maximum with increasing deformation intensity but roughly iso-
thermal conditions, the observations in nature can more easily be interpreted as
a function of total strain instead of temperature and a texture dependency on
SGR or GBM. Few studies which have a good control on strain and a similar tex-
ture evolution is observed, comparable to the texture evolution in regime 2,3
(Pennacchioni et al., 2010) which is in support of our interpretation. Examples of
B-domain textures observed at high temperatures (e.g. Menegon et al., 2011)
may not indicate an abnormal activity of (c)<a> but rather a high stress texture
forming mechanism similar to NGF as suggested here. With relation to temperat-
ure determinations, one should also note that within the presented experiments,



a range of c-axis opening angles can be found (Fig.3, 4). The observation that the
this angle seems to vary as a function of grain shape may have the potential to
add another complexity to the c-axis opening angle thermometer.
The different contributions of grain boundary sliding in each regime, in addition
to the formation of new, small grains by growth, dominantly in regime 1, may
challenge our understanding of the grains size- stress relation with respect to
piezometric applications. Similar indications could be drawn by the grain size –
gKAM relation which was documented in the companion paper (Heilbronner &
Kilian, this volume).
A method to allow the determination of the kinematic vorticity number $W_k$ are
based on the measurements of principal and oblique foliations and using the
central c-axis girdle to determine the kinematic reference frame (Wallis, 1995). It
has been suggested that those are not applicable to the experimentally deformed
BHQ based on the experiments of Heilbronner & Tullis (2006; Xypolias, 2009),
since it seems the entire girdle rotated with (local) sample strain. Although the
current observations show that the general trend of the outer part of the girdle
varies with θ (Table 1), if a sufficiently sharp kernel is used (Fig. 1, Appendix 3),
it can be seen that the central part of of the girdle remains roughly perpendicu-
lar to the shear plane. Measuring the bulk and oblique foliations in the samples
and using the method of Wallis (1995), for regime 1, the same kinematic vorticity
number is obtained as calculated from the mechanical data ($W_k$=0.96). However,
that result might be considered as a coincidence because in regime 2 ($0.76_{Wal-}$
$_{lis}$/$0.93_{bulk}$) and regime 3 ($0.99_{Wallis}$/$0.92_{bulk}$) obtained values highly deviate. Addi-
tionally, it might be questionable whether an oblique grain shape preferred foli-
ation needs to develop parallel to the ISA or if it might be influenced by the re-
crystallization mechanisms, especially in regime 1.





**5. Summary & Conclusions**
To study the textures of BHQ, deformed in general shear to high strain in re-
gimes 1, 2, and 3,  EBSD data scanned at moderate spatial resolution (0.5 and 1
µm step size), were analysed using a number of new methods for combined tex-
ture and microstructure analysis.
(1) Texture analysis was carried out for classes of different grain size, shape,
long axis alignment and grain kernel average misorientation (gKAM) and the
geometry of the pole figure skeleton varies systematically within the grain prop-
erty parameter space. Average grain properties were displayed in c-axis pole fig-
ure space to highlight how grain properties correlate with texture.
(2)  The grain kernel average misorientation (gKAM) was introduced as a meas-
ure for intragranular deformation intensity.
(3) Separate maps of texture strength as a function of grain properties (size, as-
pect ratio, long axis trend, gKAM) allow for a quantitative comparison and reveal
how the bulk texture strength and volumes of the two most prominent fibres, the
B- and Y-fibres, decrease and increase, respectively, during the transition from
regime 1 to 3.
(4) Textural neighbourhood relations are visualized on maps showing that grains
inside the Y subdomains of regime 3 grains are separated by boundaries with
higher and lower <a>-intransparency compared to a random distribution, which
can be taken as a result of subgrain rotation recrystallization with an addition of
geometrically necessary grain boundary sliding.
(5) A colouring system is introduced to connect information from misorientation
axes in the specimen and crystal coordinates. The bulk distribution of misorienta-
tion axes is always parallel to the vorticity axis, and coincident with rotations
which are dominantly around [c] for Y-grains, in R-grains the distribution of  rota-
tion axes consist of superposition compatible with a combination of misorienta-



tion axes compatible with {m}<a>, {z}<a> and {pi'}<a> derived tilt boundar-
ies. For B-grains, no conclusive preference is found, especially no clear indica-
tion of a dominance of (c)<a>.
(6) Calculation of the mean generalized Schmid factor on different <a> slip sys-
tems as a function of orientation of the applied stress with respect to the sample,
shows that in general shear experiments (c)<a> is less likely to operate com-
pared to slip systems such as {pi'}<a>.
Black Hills Quartzite deformed in general shear shows a texture dominated by
the alignment of <a> with the trend of the direction of maximum shear stress
and a transition of [c] maximum from peripheral (regime 1) to central directions
(regime 3) along a kinked single girdle (regime 2). The transition of texture geo-
metry is accompanied by an increase in texture strength. On the basis of the de-
tailed analysis presented above we propose new interpretations concerning tex-
ture formation and texture transition, as well as the relation between c-axis ori-
entation and active slip systems and that between texture and recrystallized
grain size.
Of the three dislocation creep regimes, only regime 1 and 3 are considered dis-
tinct with regime 2 being transitional. Recrystallization in regime 3 happens by
subgrain rotation recrystallization and deformation occurs though dislocation
creep with a minor contribution of grain boundary sliding. Porphyroclasts in all
regimes  deform by dislocation glide and climb.
Texture variation as a function of grain size, grain lengthening, long axis align-
ment and gKAM indicate that [c] rotates away from the periphery of the pole fig-
ure with increasing deformation intensity at the grain scale and this rotation is
accompanied with a strengthening of the texture. The increasing texture
strength is interpreted as a decreasing contribution of grain boundary sliding
and an increasing contribution of dislocation glide and climb.



Since the stress level of the experiments is the only differing parameter and
there is not observation of a rotation of [c] towards the periphery of the pole fig-
ure, we propose and additional texture forming process which operates most effi-
ciently in the high stress regime, i.e. stress controlled growth of new grains. The
texture will be a result of the balance of contributions of dislocation glide on sev-
eral <a> slip systems with [c] attracted towards a bi-maximum at the centre of
the pole figure as a function of strain and new grains formed at the periphery of
the pole figure, their amount controlled by the stress level. Grains subsequently
grow and deform by dislocation glide and grain boundary sliding. The texture
presents a dynamic balance between both processes. The hypothesis that (c)<a>
slip is responsible for peripheral [c] maxima, {m}<a> slip for the central Y-max-
imum and {pi'}/{z}<a>  for the symmetrically disposed R-maxima, is too
simplistic and our interpretation of the texture evolution opens a new field in the
interpretation of naturally produced textures. According to our interpretation, a
temperature dependency of quartz textures geometry is of indirect nature while
the relative contribution of the two texture forming processes are the controlling
factors.



**Tables**

Table 1: Variation of c-axis girdle trend within each θ class given in Figure 4. For regime 2, also the trend of the strongest maximum, located at the periphery for <a>, and {r} is given. Angles are clockwise with 0° in the West, shear sense is sinistral.

|          | θ class 1     | θ class 2     | θ class 3     | θ class4     |
|----------|---------------|---------------|---------------|--------------|
| regime 1 | 91            | 74            | 79            | 90           |
| regime 2 | 85 (-10, 121) | 67 (-18, 114) | 72 (-14, 119) | 90 (-8, 123) |
| regime 3 | 75            | 68            | 72            | 82           |





**Figure captions**
Figure 1: Pole figures of high strain samples of regime 1 to 3. (a) Pole figures for
poles to planes c,a,m,r and z from EBSD mappings of sample w1092, w946 and
w935. Pole figures oriented with the shear zone boundary (forcing blocks) hori-
zontal, shear sense is sinistral. Maximum density and pole figure J-index, given at
left and right top of each pole figure, kernel halfwidth 6°. Contour intervals at 2
times uniform density. (b) Inverse pole figures for selected reference directions:
loading direction ("$\sigma_1$"), shortening instantaneous stretching axis (ISA$_2$), major
axis direction of the ellipsoid ($\theta'$) obtained from total sample strain, shear zone
boundary (shzb), parallel to the forcing blocks in the experiments, and the plane
-45° to the extending ISA$_1$. Angles above each plot give the trend of the reference
directions, all with inclinations of 90° (in the image plane). Contour intervals at 2
times uniform density.
Figure 2: Orientation maps for samples of regime 1 to 3, w1092, w946 and w935.
(a) Inverse pole figure colour-coding using the inferred vorticity axis ( = speci-
men z direction = strain Y-direction) as a reference. colour key for purely rota-
tional point group. (b) Inverse pole figure colour-coding using the ISA$_2$-45° as a
reference direction (indicated at bottom) and a colour key for Laue point group
symmetry.  Note, for regime 2 and 3 samples, include parts of the single crystal
quartz forcing blocks at the top and bottom.
Figure 3: C-axes pole figures for different classes of aspect ratio, grain size and
$\theta$. On the left, aspect ratio increases to the right, grain size downwards and son
the right, grain long axis trend ($\theta$) in four classes clockwise as indicated by range
of angles above pole figures, and aspect ratios increasing downwards. Class lim-
its for each property are at equally spaced quantiles of the entire population.
Textures are calculated for grain modal orientations (one orientation per grain).
Maximum density and pfJ at top left and top right of each pole figure, number of



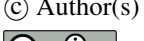

grains within each class at the bottom. Contours at 0.5 times uniform density.
Shear zone boundary horizontal, shear sense sinistral, upper hemisphere, equal
area projections. The rose diagrams at the bottom show the θ distribution and
the corresponding classes. The blue diamond indicates the direction of the long
axis of the strain ellipsoid and the red circle the direction of the ISA.
Figure 4: Pole figures <a> and {r} in regime 2. Pole figures are calculated for
different classes of aspect ratio, grain size and θ for classes given in Figure 3.
Figure 5: Pole figures of grain size, aspect ratio and axial ratio. Average grain
sizer, aspect ratio and axial ratio estimated on the c-axis pole figure for grains
smaller 25 µm. Upper hemisphere, equal area projection, greyscale bar indicates
the average grain size, aspect ratio and axial ratio. Both, aspect ratio and axial
ratio are displayed to separately visualize elongated and round grains.
Figure 6: Pole figures for low and high grain kernel average misorientation. (a)
Separate pole figures are calculated using grain modal orientations (1 orienta-
tion per grain) and all orientations (1 orientation per pixel), separately for grains
with gKAM below and above the median value. Grain size ranges of 1-12 µm (re-
gime 1,2, w1092, w946) and 1-25 µm (regime , w9353) were used. (b) Separate
pole figures for grain modal orientations and all orientations for grains with a
gKAM below the 0.2 quantile and above the 0.8 quantile. Same grain size ranges
a in (a). Maximum density and pfJ given at top left and top right of each pole fig-
ure, texture index at bottom left and number of grains or orientations within
class at the bottom right. Upper hemisphere, equal area projection, contours at 1
times uniform.





Figure 7 (landscape): Quantitative comparison of texture index and volumes of
texture components. Separate colour maps for texture index, B-fibre and Y-fibre
volume as a function of aspect ratio, grain size and θ, calculated for low and high
gKAM populations.Fibre volumes are calculated as the volume of the ODF within
a 30° radius around a c-axis fibre directed towards the peripheral (B-fibre) and
the central c-axis maximum (Y-fibre). Absolute values within each column of col-
our maps are quantitatively comparable. See text for details.
Figure 8: Slip direction intransparency and misorientation axes at boundaries
and low angle boundaries. (a) EBSD map of  sample w935 (regime 3) with colour-
coded [c] direction (Y-domain = pastel colours, peripheral [c] directions = satur-
ated colours), grain boundaries are colour-coded according to the minimal angle
between <a> (ignoring polarity) across a grain boundary. White boundaries are
transparent for <a>-slip (if adjacent glide planes are also favourably oriented).
colour coding is shown in upper hemisphere, equal area [c] pole figure of 1000
randomly selected orientations. (b) Same area as in (a) with grey value indicating
the angular distance of [c] from the periphery, and low angle boundaries (2-9°
misorientation angle) colour-coded for the misorientation axis in crystal coordin-
ates. Grain boundaries are not shown for clarity.
Figure 9: Misorientation axes of three texture domains. (a) Map with texture do-
mains determined within an angle of 25° around the orientations forming local
maxima in the ODF, colour-coded in red (Y-grains), blue (B-grains), pink (R-
grains), green (σ-grains), and yellow (all other grains). On right, from top to bot-
tom pole figures for [c], <a> and {r} showing a subset of poles. (b) colour-cod-
ing scheme used for misorientation axis directions (see main text for details). (c)
Colouring used for crystal directions, rotation axes (assuming pure tilt boundar-
ies) are indicated for some slip systems. (d) Misorientation axes obtained for low
angle boundaries (misorientation angles of 2-9°) from grains in the Y-, B- or R-
texture component. Misorientation axes are plotted with density contours at

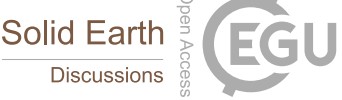



steps of 1 times uniform. Point plots are from randomly drawn subsets. Note that
directions of highest density in specimen coordinates do not always coincide with
the the direction of highest densities in crystal coordinates (e.g. for the R-
grains).
Figure 10: Average, generalized Schmid factors (Sf). Sfs are shown for different
slip systems (or combinations of slip systems) as a function of the orientation of
the stress tensor. A Sf is calculated for the modal orientations of all grains (top)
and for every orientation within a map (bottom). The mean Sf is plotted as a
function of the direction of $\sigma_1$. On the x-axis, 0° corresponds to a $\sigma_1$ direction of
45° with respect to the shear plane, negative and positive angles correspond to
synthetic or antithetic rotations of $\sigma_1$ directions.



**Appendix**
EBSD data processing
Single mis-indexed pixels have been deleted and reconstructed together with
single non-indexed pixels. Subsequently, non-indexed areas not thicker than two
pixels wide along grain boundaries were filled during noise removal using a half-
quadratic filter (Bergmann et al, 2015). The procedure was adjusted to be edge
preserving for continuous boundaries above 1.3° misorientation angle. Grains
are calculated using the segmentation algorithm implemented in the mtex tool-
box (Bachman, 2011) with a threshold of 10° boundary misorientation angle us-
ing the point group 622 and transforming the grain mean orientation back into
trigonal point group 321. This procedure avoids Dauphiné twin boundaries (60°
rotation around [c] ) being erroneously identified as grain boundaries but the
main advantage is that the mean orientation of the hexagonal grain represents
the modal orientation of the trigonal grain (representing the orientation of the
twinned domain occupying the largest area fraction of the grain). The segmenta-
tion procedure yields identical results to segmenting grain maintaining the tri-
gonal symmetry and merging grains with boundaries obeying the Dauphiné twin
relation ship within a 3° angular interval between the twinning misorientation
and the measured grain boundary misorientation (Note this angle refers to the
angle between the (mis)orientation and the twinning twinning rotation and hence
is always larger or equal to the error allowed for the twinning axis or twinning
angle). The latter procedure has the disadvantage that it will not produce a
meaningful average grain orientation, while the former is also computationally
less expensive.
Texture calculations
Contoured pole figures and inverse pole figures are calculated from the orienta-
tion distribution functions (ODF). The ODF was calculated for either all measure-
ments (area weighted) or the grain modal orientations (number weighted or in

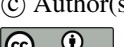


some terminology referred to as one-point-per grain). In case of ODF calcula-
tions, the de la ValeePoussin kernel was used. The kernel width was either fixed
or estimated using the Kullback-Leibler cross validation implementation in Mtex.
Estimated kernel half widths are between 7-14°.
The strength of individual pole figures is given by the maximum of the density
distribution and by the pole figure J-index (pfJ, L2-norm of the density distribu-
tion on the sphere) as defined by Mainprice et al. (2014) which is more suitable
for multi-modal distributions. A uniform pole figure will have a pfJ of 1. When
comparing pfJ values of different crystal directions the respective multiplicity
(c=2, a=3, m,r,z=6) has to be taken into account.
gKAM
In order to access how much of internal deformation in the sense of a change of
orientation within a grain is present, the misorientation to the grain mean orient-
ation can be colour-coded (Fig. A1,a,d). For this purpose, twinning is ignored and
an inverse pole figure colour-coding is chosen in such a way that the mean crys-
tal direction coincides with the mean orientation times the white centre of in the
inverse pole figure. Misorientations which relate to misorientation angles up to
30° are displayed. In order to inspect the misorientation density inside a grain,
imposed by the number and the misorientation angle of low angle boundaries, we
use a measure based on the kernel average misorientation (KAM) and the size of
the grain. Calculation of the KAM is performed using noise-reduced data (Fig. 2
b,e), since the orientation noise of conventional EBSD renders most KAM inform-
ation ambiguous. The KAM is the average misorientation angle over a kernel
computed for each measurement point. Misorientation angles above a threshold
of 8° were ignored. The grain averaged KAM (gKAM) was computed as the KAM
divided by the number of indexed pixels for each grain (Fig. 2c,f). The size of the
kernel was individually chosen to be of the order 3 or 4, which compares to a 24
or 40 pixel neighbourhood such that grains without any substructures but differ-
ent grain size maintain an identical gKAM and any grain size dependency is sup-





pressed as best as possible. In noise free data, the magnitude of the gKAM is a
measure of the low angle boundary density and hence an indirect measure for
how much grains deformed during dislocation creep. An advantage over the
grain orientation spread (GOS) is that it will not be influenced by interior high
angle boundaries or twins, however, the gKAM does not measure continuous lat-
tice bending.
Figure captions for appendix figures
Figure A1
Explanation of the gKAM: Crops of EBSD maps of 0.25 and 1 µm step size show-
ing (a,d) the misorientation to the mean orientation within a tightly confined col-
our range after noise removal. (b,e) Kernel average misorientation (KAM) of 3rd
order (24 pixel neighbourhood) of boundaries below 8° misorientation angle. (c,f)
grain averaged KAM as defined by the sum of the KAM of all pixels within a grain
divided by the number of pixels. The gkAM can be seen as a measure of misori-
entation density within a grain, imposed by the frequency and the angle of low
angle boundaries. The absolute magnitude of the gKAM will depend on the order
of the KAM and the step size and the noise level.
Figure A2
Distribution of angles between adjacent <a> directions across a grain boundar-
ies in regime 3. Histogram for grains within the Y-domains (contained within a
40° cone for the c-axis directions), out side this domain and for interdomainal
boundaries. Stippled line shows the distribution expected for a uniformly tex-
tured aggregate.
Figure A3





Pole figures of c,a,r from regime 1,2 and 3 entire sample and directions corres-
ponding to the orientation modes found in the ODF with densities >3. Numbers
correspond to the poles obtained from modes in with decreasing density. Posi-
tions of highest densities on a pole figure do not need to exactly coincide with
poles to the modes. (a,b,c) are upper hemisphere, equal area projections. Note
Bi-modal character of [c] distributions in the centre of the pole figure in regime 2
and 3.
Figure A4
Misorientation axes for regime 1, 2, and 3. colour coding and types of plots
identical to Figure 9 in the main text, except that misorientation axes for all tex-
ture components are shown.



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



Figure 1





## Figure 2

regime 1    regime 2    regime 3

(a)

reference direction perpendicular thin section plane (= specimen z)

(b)

reference direction ——— -8°
(ISA2-45°)
——— shear zone
boundary (szb)

ref. direction ——— -10°

ref. direction ——— -11°

w1092: 850 °C / "as-is" / $3*10^{-5}$ s⁻¹
$\Gamma_{effective}$=3.3*; 39% vert. shortening*
$\tau_{peak}$ = 650 MPa, $\tau$ = 315 MPa

w946: 875 °C / 0.17 wt% $H_2O$ / $3*10^{-5}$ s⁻¹
$\Gamma_{effective}$= 3.3;  48% vert. shortening
$\tau$ = 201 MPa

w935: 915 °C /  0.17 wt% $H_2O$ /$3*10^{-5}$ s⁻¹
$\Gamma_{effective}$=3.0;  46% vert. shortening
$\tau$ = 103 MPa





Figure 3




Figure 4





Figure 5

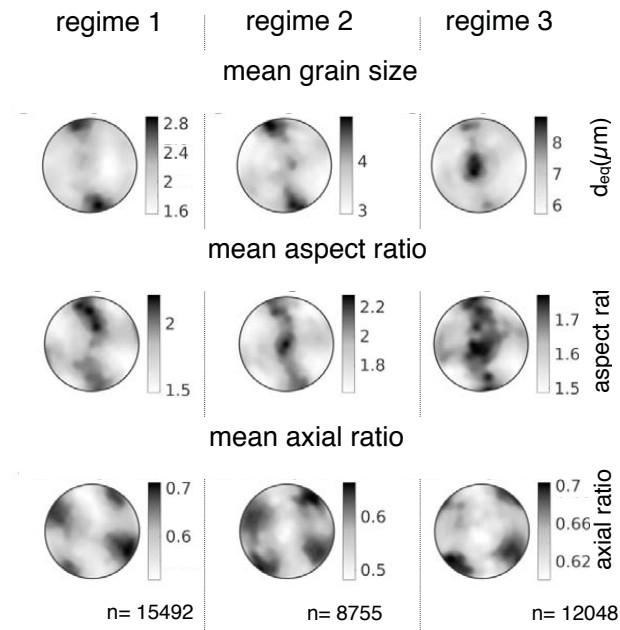



# Figure 6a





Figure 6b

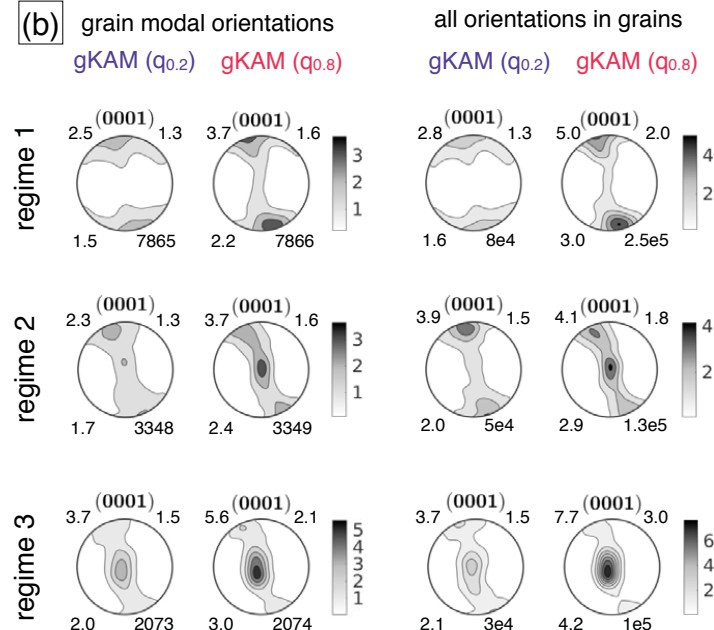




Figure 7





Figure 8





Figure 9







Figure 10





Appendix Figure A1

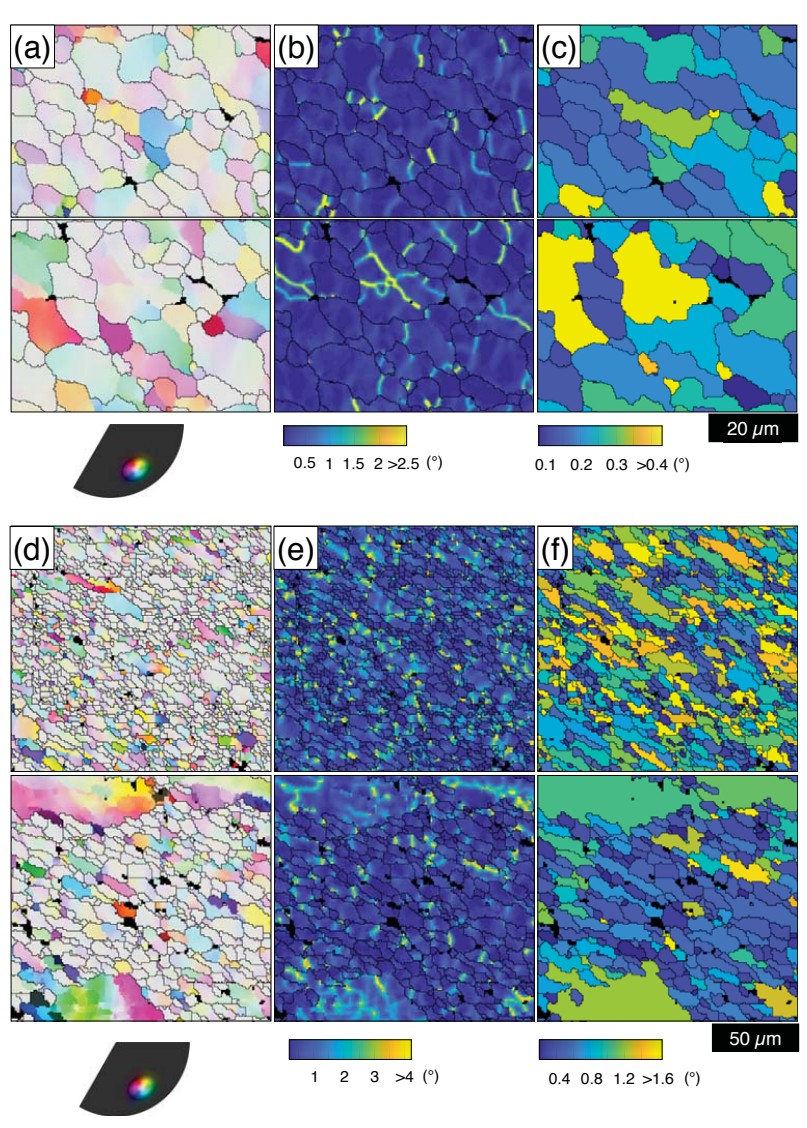





## Appendix Figure A2

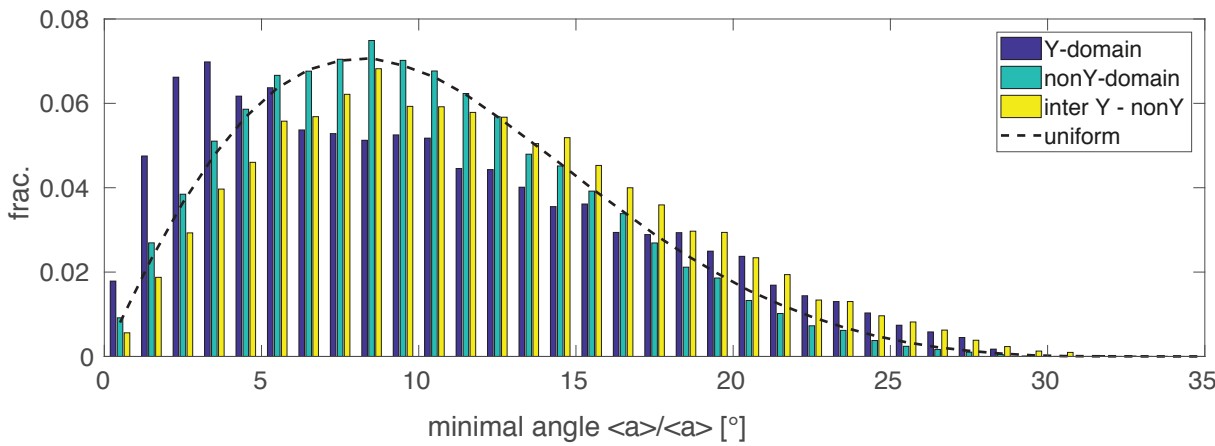





Appendix Figure A3

## Polefigures and the first n modal orientations

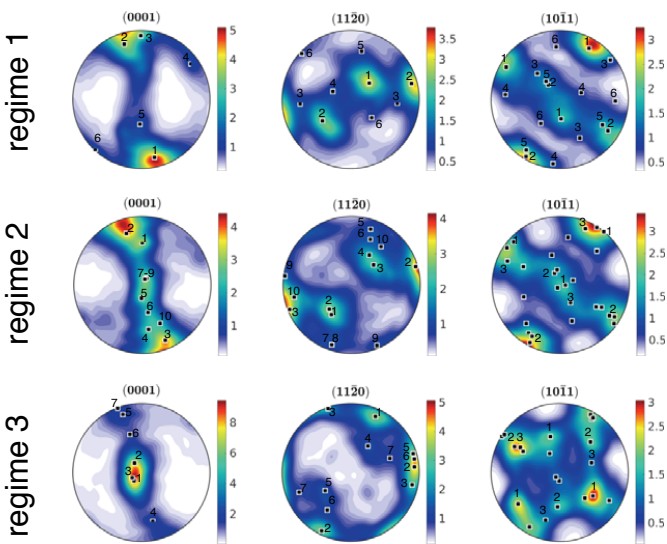



Appendix Figure A4

(a)

regime 1

(b)

misorientations axes
crystal ref. frame

misorientation axes
specimen ref. frame

(c)

y-grains

b-grains

r-grains

σ-grains

remaining

for 2-9° misorientation angle




Appendix Figure A4

regime 2

for 2-9° misorientation angle

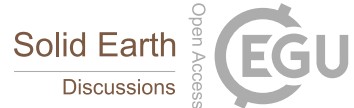

Appendix Figure A4

regime 3

for 2-9° misorientation angle