# Peer review of "Texture analysis of experimentally deformed Black Hills Quartzite"

_Solid Earth, 2017_

## Referee Comment (RC1) · L. Morales (Referee) · 29 Jun 2017

First of all I would like to apologize for the delay in my review. In this paper, the authors performed detailed microstructural and texture analyses of three samples of Black Hills quartzite deformed experimentally in the dislocation creep regime 1 to 3 (Hirth and Tullis 1992). The chosen samples were deformed at relatively high bulk strains (gamma ∼3) under similar temperatures (temperature range of 65°C) and similar constant rates, and the only difference is the peak and flow stresses. Their results show that the samples deformed in the regime 1 and 3 are clear end members in terms of microstructures and textures (in terms of CPO), while the sample from regime 2 being the transitional member. The authors shows that the CPO strength increasing towards the regime 3 is directly related to the deformation intensity at the grain scale and on the contribu-

tion to dislocation creep regime, that also increases towards this regime. The paper is provocative as it proposes that peripheral quartz [c] axes are not related to basal <a> slip but due to nucleation and growth of specific fragments during microcracking in a high stress environment. By itself this is a very important observation/conclusion, as most of the papers showing similar patterns are interpreted in terms of basal <a> slip activity and sometimes even in terms of deformation temperatures (where the presence of this patterns indicating low temperatures). The authors performed a very detailed CPO analyses in terms of grain sizes, aspect/axial ratios and even proposed a new color-coding for misorientation analyses combining the crystal and sample reference frame. The paper is well written, although here and there one may find some very long sentences that are hard to follow (I've tried to specified them below), and the figures are in general quite good and very illustrative. This paper will certainly have a high number of citations in the future and certainly fits the topics published in the Solid Earth journal. Before publication nevertheless, the authors may want to consider the following comments/suggestions:

1) The authors suggested that the quartz grains with peripheral c-axes are not due to the activity of basal <a> slip, but result from stress-controlled growth of new grains. The authors mention that they "do not have any direct evidence of fracture/nucleation/growth" (lines 654-655), but compare their observations with other experimental work where the grains in the B domain seem to be first to form under high stresses. I am quite sure the authors had a careful look for the presence of fluid inclusions related to those peripheral, but I think the authors can improve the "speculation" about the origin of those grains. For instance, Karato and Masuda (1989 – Geology) demonstrated that fine grained quartz aggregate may have significant grain growth during high temperature experiments when water is present. In their case, flatten quartz grains with aspect ratios much higher than the deformation-induced flattening were observed during recrystallization at relatively low T and high deviatoric stresses, while in higher temperatures and low atrain rates, show small grain flattening. According to their model the difference in microstructures is due to the distribution of water in

different conditions: while in low deviatoric stresses the distribution of water is more isotropic, occurring in isolated pores. Under a high stress however, the fluid phase will form a film normal to sigma 3. This roughly means that the longest axis of these films will be parallel to sigma 1. Now assuming that this is the case, and assuming that quartz will "grow" more along the c-axis (e.g. Iwasaki et al. 1997 – Journal of Crystal Growth), that might explain the peripheral grains. The authors may also want to have a look in Masuda and Fujimura (1981 – Tectonophysics) I am not sure how elasticity would influence (comments from the authors, line 662), but I think it is worth to discuss also this possibility;

2) The authors mentioned that the in certain conditions, the quartz c-axes form not one maxima around Y, but two, separated by a certain angle. This is not visible in any of the images, and I think this is an important observation. In addition, it was not clear to me why do you form two maxima instead of one, and I think this part deserves a longer discussion;

3) I am aware that the authors performed an extensive literature review about the possibility (or not) of basal <a> slip occur in quartz, but this is not really presented in the manuscript. Because this is a very important point of the paper, I believe the full review they have performed will help to clarify a number of obscure points about quartz slip systems in general, and in particular (c)<a>;

4) The color-coding linking the misorientation information from the crystal reference frame and the sample reference frame is an innovative idea here, but the bottom half of the triangle is too dark in the printed version (they are fine in the computer). Maybe this is something related to the "contrast" in the colorcode that you can change? In addition, because everything is very dark, one cannot read what is written over there, so I recommend the authors to write in white. Also, because you are using a point group 321, would not be better to have a representation of $120°$ fundamental sector, instead of $60°$

Minor comments include:

Line 25: "...not due to deformation by basal <a> slip, but possibly by..." – complete

Lines 33-42 – this sentence is too long and hard to follow, I recommend the authors to slip it in two or three shorter sentences;

Line 44 – "the underlying mechanisms and processes of texture development are poorly understood. In addition, the relationship between temperature and recrystallization mechanisms (...) or texture geometry and strain in polycrystalline materials are not always separated.

Line 53 - ...(e.g. Blacic, 1975). However, conclusions...

Line 70 – ...(stipp and kunze, 2008). In this sample, recrystallization...

Line 102 – Pt jacket

Line 113 – Forgot to insert the details here

Line 116 – why sample w1092 appear twice?

Line 126 – Crystal directions [0001],<11-20> and poles to planes (0001), ...

Line 134 – You mentioned here that "grains from different maps of identical stepsize...have been combined", you should add that the maps belong to the same sample;

Line 141 – "because of...", instead of "since";

Line 172 – can you clarify why you have used a misorientation of 9°, instead of 10 or 15°?

Line 226 – A schematic inset showing the directions for the IPF would be very helpful, particularly for those readers starting to work with CPOs;

Line 262 – Somewhere in this paragraph you have to state that theta is the angle

between the shear direction and the longest axes of quartz, this is not clear here;

Line 287 – Move this paragraph after Line 280, it makes more sense there;

Line 290 – It might be worth writing a sentence explaining what is the difference between aspect ratio and axial ratio;

Line 318 and Figure 7 – this figure needs a better explanation and a better labelling, it is not easy for a non-specialist to read these 2D histograms. In addition the axes in the graphics are not very clear

Line 350 – I suggest the authors to add one or two lines about the <a>-intransparency;

Line 374 – Where is the <7-2-56> in the IPFs?

Line 391 – "independent", instead of "disrespect"

Line 400 – {m} does not have this crystallographic. . .

Line 494 – You may want to cite Hansen et al. 2012 JGR?;

Line 503 – 507 – this sentence is a bit confuse, I suggest the authors to rewrite it;

Lines 509 – 512 – You could possibly quantify that the newly formed grains in regime 1 are smaller and have less lattice distortion, meaning that they are less deformed, this seems to be the case at least for the grain sizes from Heilbronner and Kilian, the companion paper of this manuscript;

Line 556 – {a} or {m}?

Line 592 – 597 – sentence too long, please split in 2;

Line 598 – 599 – this sentence is confusing. . .did you mean " Because the displacement rate is constant and the temperature differences are very small in the current experiments. . ."

Line 606 – as well as is repeated;

Lines 616 – 645 – I found this part a bit confusing, maybe the authors want to rephrase some of the sentences and clarify the text;

Line 669 – references needed after "...as an easy slip system." – you may introduce them here if you consider my third comment;

Figures – I recommend the authors to increase the font sizes, particularly from the figures with a number of different pole figures. Also, I suggest the authors to use the 'contourf' option in their plots of Figure 5. I think a simple sketch showing the reference frame of the pole figures and illustrating the main directions used for the IPFs (in relation to the experiment assembly) will be very helpful

---

## Referee Comment (RC2) · D. Prior (Referee) · 7 Aug 2017

Reviewer's comments on "Texture analysis of experimentally deformed Black Hills Quartzite" by Rüdiger Kilian and Renée Heilbronner. Submitted to Solid Earth.
Review by Dave Prior, University of Otago

Apologies for being slow in reviewing.

This paper has the potential to make an excellent contribution to our understanding of quartz deformation. The paper presents a high quality EBSD data set from some experimentally sheared quartzites. The data are used to demonstrate that there is a transition in crystallographic preferred orientations (CPO) as a function of different deformation conditions. In a high stress magnitude experiment [c] axes are clustered around a direction rotated a few degrees away from the normal to the shear plane. In a low stress experiment the [c] axes are clustered in the shear plane normal to the shear direction. The microstructures of these experiments correspond to recrystallisation regimes 1 (high stress) and 3 (low stress) as defined by (Hirth and Tullis, 1992). CPOs from a regime 2 (intermediate stress) sample are transitional. The authors use the EBSD data to relate closely the CPO and the sample microstructure to with the aim of providing a more robust understanding of CPO forming processes. The authors argue that long held ideas of the transition they describe being related to changes in dominant slip system as a function of increasing temperature are not consistent with the new data. One key interpretive outcome is that basal <a> is not an important slip system in quartz deformation. If this is demonstrable, it is really important and upsets conventional thinking. This work has specific importance to researchers using quartz microstructures to understand crustal tectonics and more general importance to researchers with an interest in deformation microstructures.

Although I think this is an important paper, it needs major modifications to ensure that it has appropriate impact. In its current form it will be ignored and very few will understand it. At present it is way too long and it's a really hard read. It needs to be rewritten so that it is easier to understand by both a general audience and by microstructural specialists. I consider myself a microstructural specialist and there are large parts of the manuscript that are completely indecipherable. There are sophisticated analyses that do not really contribute to the paper's key conclusions and these need to be reduced or removed to focus on the key new information. The discussion and conclusions do not build upon the observations in a logical way that is easy to follow. A much shorter more focussed discussion will be much more effective. I think that it is possible to reduce the paper to a much better paper of one third to one half the current length. The paper is sufficiently unclear that I cannot really judge the robustness of the interpretations. The paper needs to make its case with considerably more clarity.

Below I outline some scientific points I think the authors should address and a set of recommendations on how to make this paper much more accessible. What I've written here is a mere subset of more extensive comments on an annotated .pdf file of the manuscript. In is important to bear in mind that Solid Earth is a

broad journal and it is important that geophysicists, petrologists etc read this paper and understand it. In it's current form they will likely not read it and will find it very difficult to understand.

**Scientific comments.**

i. *CPOs a function of stress.* The authors argue that the three samples are all deformed at about the same temperature and at the same strain rate and that therefore the stress is the key difference between the samples. Although I agree that for these experiments stress is the parameter that changes the most, I do not think that allows you to argue that temperature or strain rate cannot also be parameters that will affect the microstructures. (Hirth and Tullis, 1992) relate the three regimes to both temperature and strain rate and could equally well have related them to stress. Furthermore you need to explain why the stresses are different in the three samples. The quartzite behaviour will be described by some composite flow law, so changing stress in isolation is highly unlikely; different strain-rate, T, or water content must control this. Although the temperatures are argued to be almost the same, the change in temperatures across the three samples is consistent with the changes in stress (lower T gives higher stress). If I plot a log stress vs 1/T Arhenius type plot using the flow stresses I get a straight line for all three points although the activation enthalpy I calculate from this (760kJ/mol) is much higher than literature values (Hirth et al., 2001). This makes me wonder whether anything else is changing. The key candidate for me is strain rate. The authors say the samples are all deformed at the same rate. The rate is quoted in the methods as the axial shortening rate*, which I would presume is that measured from piston displacements during the experiments. Given that these are general shear experiments I think it's highly unlikely the rates are the same for all experiments. Indeed the table in the companion paper lists them as different. My conclusion is that the flow stresses you have are the function of T and strain rate variations between the samples. The authors make some comments on the role of water from line 592 (too late in paper). Maybe water content differences explain the stress differences (although two samples are listed as the same water added?). Even if this is the case arguing stress as the only parameter is naïve.

* note: I cannot make sense of the strain rates. I don't know where sample w1092 is from. It is not used in either of the Heilbronner & Tullis papers quoted. w935 is assigned a shear strain rate of $2*10^{-5}s^{-1}$ by Heilbronner and Tullis 2006 and the methods say that an axial shortening rate of $1.2 *10^{-6}s^{-1}$ was used. For the same sample (and for w946) Heilbronner and Tullis 2002 provides a shear strain rate of $3*10^{-5}s^{-1}$ (as used in this manuscript). I've just looked at table one in the companion manuscript (H&T 2017- that gives a range of shear strain rates for each of the three samples and they are not the same. It's not very good if two papers on the same samples in the same journal and year have different values for strain rate! I'm sure there is an explanation- important to sort this out.

ii. *Lack of strain series*. The three samples are all deformed to about the same strain and probably along different mechanical and microstructural pathways. The strain series has two points each (undeformed and deformed) on three pathways. In this case, the coincidence of CPO components (e.g. Y maximum) with particular microstructures (e.g. high intragranular strain) is not documentation of a kinematic model (e.g. rotation of c-axes from the

periphery). Similarly changes from regime 1 to regime 3 are not representative of a strain series.  It's not robust to make interpretations of CPO evolution from your data alone. The discussion of kinematic models needs to bring in the evidence from experiments or natural samples where a strain/ time series is reasonable (e.g. (Cross et al., 2017; Heilbronner and Tullis, 2006) from experiments, several papers from natural examples).

iii.    *<a> axis alignment.*  I really like the idea of using the intergrain <a> axis alignment (fig 8). I think this general approach could be very useful in other systems (ice in particular). I have a gut feeling that something useful can come from this analysis for this paper. However, at present the use of these data fall into the class of "fancy analysis that goes nowhere" in point 6 below. Part of the problem maybe that you show this only for the regime 3 sample. All the other key points in this paper come from comparison of the three samples and I think this is necessary to show the value of the <a> axis alignment. For example is the pattern shown in fig A2 the same in all three regimes? I think that a key point that may come from fig 8 is that the Y domains provide greater volumes (larger domains?) that can deform by <a> slip? The figure does not really show this well. I wondered whether it would be useful to show a map where domains connected by thresholds in <a>transparency are shown (and coloured as Y, B or other). This approach has some similarities to the grain boundary hierarchy approach (Trimby et al., 1998). Such data for the three regimes may show key differences in size and connectivity of <a>transparent domains. These observations may then be linked to the interpretations about slip system activity.

iv.    *Lots more*- see pdf.

**Making this paper more accessible and understandable.**

1.    *Abandon "Texture".* Throughout this paper the term "texture" is used with the meaning common in metallurgy and materials science. There is a very small community of geoscientists who use "texture" in this way. For the vast majority of the geoscience community "texture" means the spatial relationships of phases and microstructures. To most geoscientists texture is what you would see down a microscope (in a petrographic examination for example) and is broadly synonymous with the term microstructure. Textbooks in geoscience use the term texture in this way and there are specialised textbooks on textures that take this meaning (Metamorphic:, Barker, Shelley. Spry, Yardley & Mackenzie. Igneous Mackenzie & Mackenzie, McPhie et al., . Sedimentary: Holt, Scolle. Ore Minerals: Edwards, Taylor). I did a web of science search for "texture" in the title and "rock*" in the topic. This recovers 888 papers. All but 4 of the newest 50, all but 2 of the oldest 50 and all but 3 of the highest-cited 50 use texture in the geoscience way rather than the metallurgical way (I've no time to look at 888). Having "texture" in the title is particularly problematic as most geologists/ petrologists will misinterpret it and most geophysicists won't know what it is. The terms "crystallographic preferred orientation" (CPO) or "lattice preferred orientation" (LPO) are much better as they are explicit. If you want this paper (and the companion paper) to have wider readership, remove the word texture throughout and replace with CPO or LPO.

2. *A diagram to explain your reference frames.* The names used for various orientations of c-axes on the pole figures are explained in words around lines 80-93. This is not very satisfactory: these need to be shown in a figure. A figure will act as a constant point of reference for readers looking at the results or discussion and trying to remember what particular grains or domains are. Remember also that most of Solid Earth readers will be unfamiliar with the general shear kinematics. Many will be unfamiliar with pole figures and especially with inverse pole figures. So a simple figure that has:
    a. A cartoon of the general shear geometry and kinematics (like lower part of fig 1c in H&T, 2002)
    b. A stereonet to show the various "domain" and "grain" orientations used in to describe the CPOs and the angle conventions (where domains relate to variable angles). Some key directions should be highlighted on a.
    c. An IPF as used later in figs to show key crystal directions used.
    d. This figure could also be used to relate names used in this paper to alternative names used in the literature – reducing text length and making these links clearer.

    This figure will make the paper much easier to understand and will enable some of the figs that follow to be simplified.
3. *Crystal axes.* On lines 126-128 the indices and names of crystal axes are related. In the rest of the paper these two forms are interchanged. For example most of the figs use the indices and most of the text the names. This makes it difficult to read for those not familiar with indices in the trigonal system. I would stick to the names ([c], <a> etc). The words you have (l126-128) are still needs but then stick to names in text and figs.
4. *One point per grain vs one point per pixel.* There are valid reasons to explore which of these choices are made. The complications added by showing a mixture of both makes this paper harder to follow. The choice makes no significant difference in any of the data presented (this is said somewhere but can't find it) so just pick one way. This will shorten the text and reduce the size/ number of figures. I would pick one point per pixel (all orientations).
5. *IPF ref frame choice.* A mixture of extended and folded IPF forms are used. I'm sure there is an argument as to which is used where, but this is another thing that will act as a significant point of confusion. On figs 2b and 8b I can see that it is important to have the folded form. In none of the other figures does it matter. The IPFs in fig 1b are mostly symmetric and little useful information is lost in folding them (also inconsistent to show these in extended form and not <a> pole figs on whole sphere). Actually I don't think there is any information in the IPFs in Fig 1 that you don't get from the pole figs. So why not just remove them. I would switch to folded IPF throughout and also make the colour schemes of IPF s the same (e.g figs 2a,b and 8b).
6. *Fancy analysis that goes nowhere.* There is some very sophisticated analysis of the EBSD data in this paper. I'm sure it took a lot of time to work out and enact. Some of it really does not add anything (I cannot see any impact on the conclusions) and just makes the paper much harder to understand. In particular I see no value in figures 5, 7 and the two central columns of figure 9/A4 and the text that goes with these figs or figure elements. I have

commented on fig 8 earlier. The key scientific points from Figs 9 and A4 are best made by the first and fourth columns alone. Indeed I made a composite of A4 to try and understand better the data myself. I've attached my simplified fig 9 to this review and I suggest a single figure of this form will help you explain the misorientation data and it's significance much more clearly. Similarly I made a compacted version of fig 6 (also attached) to understand better what these data show. The Solid Earth readership is not the place to explore the much more sophisticated analysis I have suggested you cut (see annotated pdf). It will take a lot of work to make these approaches intelligible and further work (and space) to make it clear what scientific value they add. If you can demonstrate the value of these approach I think you should publish them – but in a journal for a more specialist audience.

7. *Describe the "plots" at the point where they are used*. The methods section includes the descriptions of the plotting approaches. When reading the results one has to constantly refer back to the methods. I would use the methods for the experiments and EBSD acquisition and processing (bring out of appendix) and describe the plots where they are presented in results. I think this will make the paper clearer and shorter.

8. *Discussion: needs a figure and shortening.* The discussion attempts to present a model to explain the CPO transitions the authors have documented. The effectiveness of the discussion will be improved by
   a. Having a cartoon figure that reminds the reader of the key observations and the presents the model built open these (much like a graphic abstract).
   b. Shortening  (a lot) and tightening. The discussion is way too long. A paper can only make so many key points effectively and this discussion is far too expansive. The effect is the reader will take very little away from the paper. I have drafted what I think are key data points from my reading and suggest that these form the framework of discussion.

9. *Make real conclusions.* The conclusions do not really give the reader any useful idea of what you have found out. They are a mess.

10. *Fold the appendices into the main text.* I don't think you have anything that needs go into an appendix- it is all better in the main text – especially if number of figures reduced.

11. *Simple writing.* The writing style is verbose. Sentences are too long. Complex phrases are used where simple ones would work better. I have highlighted examples on the annotated pdf.  Overall the writing lacks clarity and needs considerable improvement.

12. *Literature interpretation versus observations.* The discussion of the literature (particularly in the introduction and discussion) tends to focus on the interpretations of particular papers, sometimes without explicit reference to the data or observation that leads to the interpretation. For completeness and to enable better comparison to your new data literature citation needs to refer to both observations and interpretations.

Here's some key data points I get from the paper. I started writings this as a framework for my own understanding (because I could not extract this easily from reading the paper) and as a potential replacement for your conclusions. What I have here is too expansive for conclusions, although it could be a framework for the discussion. If data (figs etc) do not have a direct line through to conclusions they should be omitted.

Firstly some simple factual conclusions- should not ever be wrong. 2 to 5 could be put as one. A summary figure could schematize these characteristics

A. CPOS and microstructures, from EBSD data, are presented for three experimentally deformed quartzite samples, sheared to similar high strains in regimes 1, 2 and 3.

B. The samples show a transition in CPO. In regime 1 [c] axes are clustered around a direction (B-fibre) rotated a few degrees away from the normal to the shear plane. In regime 3 the [c] axes are clustered in the shear plane and normal to the shear direction (Y-Fibre: parallel to vorticity axis). The CPO of the regime 2 sample is transitional.

C. In all regimes the most significant <a> cluster is in the plane that contains the shear direction and the shear plane normal. The cluster lies at an angle of ~10 degrees from the shear plane, with a rotation sense compatible with vorticity.

D. The {r} rhombs are clustered around an orientation close to the maximum principal stress direction in regimes 1 and 2 and have a much weaker alignment in regime 3.

E. CPO strength increases from regime 1 to regime 3.

Then on to some more sophisticated observations. Still factual. These are more subtle points that I think are quite new and maybe important in interp. Again inclusion in a schematic diagram will help here.

F. In all three regimes CPO strength is higher in grains with higher intragranular deformation intensity as measured using a grain kernel average misorientation (gKAM). In all three regimes the ratio of the number of Y domain to B domain grains increases with intragranular deformation.

G. CPOs have been generated for data subsets with limited ranges of grain size, aspect ratio and long axis orientation. These show that strongest CPOs are developed in populations of grains with the highest aspect ratios and with long axes aligned close to the orientation of maximum elongation. These statements apply to all three regimes.

H. CPO strength increases with increasing grain size in all three regimes although this effect is stronger in regime 1 than regime 3.

I. Low angle misorientations within all grains lie dominantly parallel to the vorticity axis. Misorientations in the crystal reference frame depend on the grain orientation. Misorientation axes in Y domain grains cluster around [c], whilst those in B domain grains lie dominantly in the basal plane, with concentration around {m}. These observations are generally true of all regimes.

J. Something about <a>transparency if a useful point comes from this
K. Something about Schmid factor analysis (see fig with comparison to Little et al data)

Then onto interpretations. I am bit lost here as I'm not really clear what are your key conclusions. The conclusions in the paper certainly do not tell me. Needs to be structured as
L. Interpretations made from your data alone
M. Interpretations that bring in outside information (e.g. strain series information)

I hope this review is useful.

Refs cited in review and listed on annotated pdf.
(Kunz et al., 2009; Little et al., 2016; Pehl and Wenk, 2005; Qi et al., 2017)

Cross, A., Hirth, G., and Prior, D. J., 2017, CPO evolution: effects of secondary phases and grain boundary sliding: Geology, v. doi:10.1130/G38936.1

Heilbronner, R., and Tullis, J., 2006, Evolution of c axis pole figures and grain size during dynamic recrystallization: Results from experimentally sheared quartzite: Journal Of Geophysical Research-Solid Earth, v. 111, no. B10.

Hirth, G., Teyssier, C., and Dunlap, W. J., 2001, An evaluation of quartzite flow laws based on comparisons between experimentally and naturally deformed rocks: International Journal of Earth Sciences, v. 90, no. 1, p. 77-87.

Hirth, G., and Tullis, J., 1992, Dislocation Creep Regimes in Quartz Aggregates: Journal of Structural Geology, v. 14, no. 2, p. 145-159.

Kunz, M., Chen, K., Tamura, N., and Wenk, H. R., 2009, Evidence for residual elastic strain in deformed natural quartz: American Mineralogist, v. 94, no. 7, p. 1059-1062.

Little, T. A., Prior, D. J., and Toy, V. G., 2016, Are quartz LPOs predictably oriented with respect to the shear zone boundary?: A test from the Alpine Fault mylonites, New Zealand: Geochemistry Geophysics Geosystems, v. 17, no. 3, p. 981-999.

Pehl, J., and Wenk, H. R., 2005, Evidence for regional Dauphine twinning in quartz from the Santa Rosa mylonite zone in Southern California. A neutron diffraction study: Journal of Structural Geology, v. 27, no. 10, p. 1741-1749.

Qi, C., Goldsby, D. L., and Prior, D. J., 2017, The down-stress transition from cluster to cone fabrics in experimentally deformed ice: Earth and Planetary Science Letters, v. 471, p. 136-147.

Trimby, P. W., Prior, D. J., and Wheeler, J., 1998, Grain boundary hierarchy development in a quartz mylonite: Journal of Structural Geology, v. 20, no. 7, p. 917-935.

[Figure]

if all contoured with same intervals fewer legends needed. Also the grey scale variation does not show the casual reader that the y grain misorientations are all much stronger. Also do each of these data sets contain a comparable number of grains? Or comparable number of point pairs. Caption says drawn from randomly drawn subsets. This does not define what you have done clearly.
I would include the number of grains in each figure pair
Fromthe PFs in fig 1 I do not expect many grains in the sigma orientation and wonder whether the less organised patterns in reg 2 and 3 relate to this.  Similarly
I predict the number of b-grrains to reduce from regime 1 to 3 (from PFs)
and wonder whether some of the shape changes here reflect that?

[Figure]

Essential elements of Fig 6 needed. Does not need other directions. I don't think the microstructure pic needed but could be. This gets over the transition from B to Y with increasing intragranular strain.

**regime1**

[Figure]

Legend:
- {10-10}<1-210> prism-a
- {0001}<1-210> basal-a
- {10-11}<1-210> r rhomb-a
- {01-11}<1-210> z rhomb-a
- {10-12}<1-210> pi rhomb-a
- {01-12}<1-210> pi' rhomb-a
- z rhomb-a + pi' rhomb-a
- basal-a + prism-a
- prism-a + z rhomb-a + pi' rhomb-a

σ₁ direction

**regime3**

σ₁ direction

[Figure]

More than we published: from my data repository. Sorry labels did not translate

- basal<a>
- z<a>
- r<a>
- prism<a>
- basal & Rhombs
- All

[Figure]

a)

Z

σ₁

θ

foliation

28°

Shearbands (C')

σ₃

b) SCZ02 map

% of Schmid factors > 0.4

(z)<a>

(c)<a>

θ

c) basal + rhombs

- SCZ02 map A
- ST11A map

θ

[Figure]

**Fabric/Kinematic Reference Frame**

b)

Z {r}{z}

{r}{z}

shear band slip direction and <a> maximum in quartz

[c] girdle

σ₃

σ₂

<a>

C' plane

<a>

Lin

foliation

X

SZB

σ₂σ₃

<a>

σ₁

σ₁

{r}{z}

{r}{z}

---

## Author Comment (AC1) · 21 Aug 2017

Replies to the reviewers comments on "Texture analysis of experimentally deformed Black Hills Quartzite "

We thank both reviewers for their detailed comments and their careful study of the manuscript In the following we will answer the comments and indicate which changes were conducted to the revised version of the manuscript.

Replies and action taken wrt to reviewer 1:

Reviewers comment: *1) The authors suggested that the quartz grains with peripheral c-axes are not due to the activity of basal <a> slip, but result from stress-controlled growth of new grains. The authors mention that they "do not have any direct evidence of frac- ture/nucleation/growth" (lines 654-655), but compare their observations with other ex- perimental work where the grains in the B domain seem to be first to form under high stresses. I am quite sure the authors had a careful look for the presence of fluid inclu- sions related to those peripheral, but I think the authors can improve the "speculation" about the origin of those grains. For instance, Karato and Masuda (1989 – Geology) demonstrated that fine grained quartz aggregate may have significant grain growth dur- ing high temperature experiments when water is present. In their case, flatten quartz grains with aspect ratios much higher than the deformation-induced flattening were ob- served during recrystallization at relatively low T and high deviatoric stresses, while in higher temperatures and low atrain rates, show small grain flattening. According to their model the difference in microstructures is due to the distribution of water in different conditions: while in low deviatoric stresses the distribution of water is more isotropic, occurring in isolated pores. Under a high stress however, the fluid phase will form a film normal to sigma 3. This roughly means that the longest axis of these films will be parallel to sigma 1. Now assuming that this is the case, and assuming that quartz will "grow" more along the c-axis (e.g. Iwasaki et al. 1997 – Journal of Crystal Growth), that might explain the peripheral grains. The authors may also want to have a look in Masuda and Fujimura (1981 – Tectonophysics) I am not sure how elasticity would influence (comments from the authors, line 662), but I think it is worth to discuss also this possibility;*

our reply:  This observation and the inference coming as a result of the analysis are mainly based on texture analysis. Bringing any further data into this ms, might be too much. EBSD data to be suitable for such an in-depth exploration is not easy to obtain for regime 1 due to the high defect density in those samples. Additionally, a study with a focus on microstructure analysis is still ongoing and for now beyond the scope of this manuscript. We agree that the current manuscript will give a stimulation and reason to further investigate this issue in much more detail. With respect to the suggestion whether anisotropic grain growth under a deviatoric stress might be responsible for the observed texture, we agree that our data could theoretically fit the model of Karato & Masuda (1989). However, one has to note that the situation here is different from the novaculite growth experiments by Masuda & Fujimura (1981) and the interpretation of Karato & Masuda (1989). As-is BHQ has a water content of ~500 ppm H (Stipp et al., 2006) while novaculite contains about 200000 ppm H, which can be considered a significant difference in both materials. Secondly, without  discrediting Karato& Masuda, 1989, their

model is rather speculative and not a demonstration of anisotropic grain growth relying heavily on the basic assumption that there is no strain localization in their samples. It is known that during - especially high stress- experiments in the Griggs apparatus, fracturing is a widespread process and we interpret e.g. the green bands (former Fig. 2b, now Fig. 4b) as grains grown in extensional/opening shear bands. We think either anisotropic growth and/or an oriented nucleation are the most likely candidates to explain the observed preferred orientation, however we do not yet have a definite answer which one exactly to blame. Given that the small and rounded grains in regime 1 occur in a position consistent with c-axes parallel to the instantaneous stretching direction (high axial ratio in conjugate positions) a stress related growth mechanism might be a feasible candidate to consider.
We updated the text on the relevant potions to reflect some of those speculation.

Reviewers comment: *2) The authors mentioned that the in certain conditions, the quartz c-axes form not one maxima around Y, but two, separated by a certain angle. This is not visible in any of the images, and I think this is an important observation. In addition, it was not clear to me why do you form two maxima instead of one, and I think this part deserves a longer discussion;*

our reply: This was indeed only clearly visible in the former Appendix Figure C1 while in most other pole figures, the estimated kernel width is sufficiently large to form an elongated central maximum. The presence of the double maximum is outlined in the companion paper, but can also be seen in the literature (e.g. Pennacchioni et al., 2010). We could speculate on the reason, one likely possibility is that in the case of a strain related rotation of the c-axis direction along a girdle trajectory towards the center, that similarly with strain, rotation rate asymptotically decreases (similar to the long axis of a strain ellipsoid which can never cross but only approach the shear plane). Additionally it might be a balanced position in on {m}<a> will result in local strain incompatibilities or hardening because the slip plane rotates out of the maximum Schmid factor position, respectively cannot be aligned with neighbouring grains. Acordingly, slip in <a> direction on one of the rhomb planes might concur with {m}<a>, holding [c] in an intermediate position. Similarly, models for this rotation have been suggested by Urai et al., 1986 (Bouchez, 1976) or Ord & Hobbs (2011). Within the scope of shortening the ms, we removed any notion on the bi-modal maximum and will keep it for further investigation in the following years. Note that the companion paper clearly demonstrates the existence of 2 submaxima.

reviewers comment: *3) I am aware that the authors performed an extensive literature review about the pos- sibility (or not) of basal <a> slip occur in quartz, but this is not really presented in the manuscript. Because this is a very important point of the paper, I believe the full review they have performed will help to clarify a number of obscure points about quartz slip systems in general, and in particular (c)<a>;*

our reply:  IWe did a detailed search of the origin of the frequent assumption that (c)<a> is a) a common slip system in quartz, b) deformation is accommodated through it's activity and is accordingly for the alignment of c-axes perpendicular to the flow plane. We added an excerpt from this survey,

however we feel it is beyond the scope and also the length of this manuscript to do an in-depth explanation of those findings. We added however, that the belief in (c)<a> stems from a series of interpretations, starting off with deformation lamellae which were wrongly interpreted to represent slip planes followed by erroneously cited AGU abstracts in the 60s, citations of publications that never determined (c)<a> and citations of publications which explicitly mentioned that (c)<a> was not found but still were nevertheless cited as reference for (c)<a> being popular slip systems. Fairly recent reviews on dislocations in minerals even cite literature for the determination of (c)<a> and the temperature dependency of <a>-slip systems which was never accomplished in the cited works. However, we don't feel like pointing fingers towards some specific contribution with which the general acceptance of an efficient (c)<a> started off with, but will only notice that in the literature, efficient (c)<a> is either never observed or rather indications for it's absence were found.

reviewers comment: *4) The color-coding linking the misorientation information from the crystal reference frame and the sample reference frame is an innovative idea here, but the bottom half of the triangle is too dark in the printed version (they are fine in the computer). Maybe this is something related to the "contrast" in the colorcode that you can change? In addition, because everything is very dark, one cannot read what is written over there, so I recommend the authors to write in white. Also, because you are using a point group 321, would not be better to have a representation of 120◦ fundamental sector, instead of 60◦*

our reply: (a) The colorprofile embedded for the figure has been updated, and the colorcoding lightened. (b) With respect to colorcoding misorientation axes in crystal coordinates for a mineral with a point group 321, one could consider the following: The crystal direction representing the misorientation axis is vector in crystal coordinates around which crystal orientation A can be rotated into crystal orientation B, or rotating orientation B into orientation A. The choice of the reference system (crystal A or B) would determine the direction of this vector (polarity). Because there is no physical reason we are aware of to distinguish a rotation of A into B or B into A crystal coordinates, the rotation vector can be treated as an axis without polarity. Therefore it follows that all axes that rotate A into B or B into A, each with a symmetry of 321, can actually be represented within the point group -3m.

The reviewers minor comments have all been considered and corrected if they still apply to the revised form of the manuscript. The fontsize of the figues was increased where possible, additional explanations provided for the chosen misorientaion angles, brackets, figure annotations and citations were updated and the text has been decluttered.

Replies and action taken wrt to reviewer 2:
reviewers comment: i. *CPOs as a function of stress: CPOs a function of stress. The authors argue that the three samples are all deformed at about the same temperature and at the same strain rate and that therefore the stress is the key difference between the samples. Although I agree that for these experiments stress is the parameter that changes the most, I do not think that allows you to argue that*

*temperature or strain rate cannot also be parameters that will affect the microstructures. (Hirth and Tullis, 1992) relate the three regimes to both temperature and strain rate and could equally well have related them to stress. Furthermore you need to explain why the stresses are different in the three samples. The quartzite behaviour will be described by some composite flow law, so changing stress in isolation is highly unlikely; different strain-rate, T, or water content must control this. Although the temperatures are argued to be almost the same, the change in temperatures across the three samples is consistent with the changes in stress (lower T gives higher stress). If I plot a log stress vs 1/T Arhenius type plot using the flow stresses I get a straight line for all three points although the activation enthalpy I calculate from this (760kJ/mol) is much higher than literature values (Hirth et al., 2001). This makes me wonder whether anything else is changing. The key candidate for me is strain rate. The authors say the samples are all deformed at the same rate. The rate is quoted in the methods as the axial shortening rate\*, which I would presume is that measured from piston displacements during the experiments. Given that these are general shear experiments I think it's highly unlikely the rates are the same for all experiments. Indeed the table in the companion paper lists them as different. My conclusion is that the flow stresses you have are the function of T and strain rate variations between the samples. The authors make some comments on the role of water from line 592 (too late in paper). Maybe water content differences explain the stress differences (although two samples are listed as the same water added?). Even if this is the case arguing stress as the only parameter is naïve.*

our reply: The difference in measured sample strength stems mainly from the different amount of water and the temperature difference. The displacement rates as measured on the load piston are identical and the calculated maximum shear strain rates are in the range 2.4 e-5 to 3.1 e-5 which can be regarded as identical. The temperature range of 65° alone is not likely to explain the observed behaviour, however with the addition of water, it is obviously just sufficient to produce the different stress regimes. We updated the text to clarify that it becomes more clear that the additions of water controls the flow stress of the sample.

We added a table with the conditions and mechanical data of the experiments for a quicker overview. We also corrected the confusion with axial shortening rate/ shear strain rate. Differing numbers between Heilbronner & Tullis, 2006 and the current data is a different, more correct procedure stress/strain calculation from the experimental data, added the origin of w1092. The notion that water is the key factor controlling sample strength was moved to an earlier position in the ms and we clarified how he different flow stresses arise while the bulk strain rate is approximately constant and the temperature variation is in a negligible range.

Note1: Calculating an activation energy for a very narrow temperature range (most likely  with a very large error), provides the rather exotic result which can be taken as another indication that the two end member regimes do not share the same rate limiting process; there is no common line of activation energy to fit between water added and as-is samples, i.e. regime 1 and regime 3.

Note 2: Strain rates are clearly bulk strain rates and one might speculate about a extent of strain localization, mechanical boundary conditions and the state of stress inside those experiments, however

that is again beyond the scope of this contribution.

*reviewers comment: ii. Lack of strain series. The three samples are all deformed to about the same strain and probably along different mechanical and microstructural pathways. The strain series has two points each (undeformed and deformed) on three pathways. In this case, the coincidence of CPO components (e.g. Y maximum) with particular microstructures (e.g. high intragranular strain) is not documentation of a kinematic model (e.g. rotation of c-axes from the periphery). Similarly changes from regime 1 to regime 3 are not representative of a strain series. It's not robust to make interpretations of CPO evolution from your data alone. The discussion of kinematic models needs to bring in the evidence from experiments or natural samples where a strain/ time series is reasonable (e.g. (Cross et al., 2017; Heilbronner and Tullis, 2006) from experiments, several papers from natural examples).*

our reply: For clarification that our observation is not only restricted to intra-sample strain variation, we added some data on a low strain regime 3 sample.
We agree that the wording CPO evolution is misleading since it indeed does not represent an evolution with strain since we cannot track one and the same grain through deformation. However, since a deformed grain must once have been an undeformed grain, our analysis can still be regarded as valid and representative of an evolution of a texture property with deformation. Notably, running these sort of experiments only to low strain does not necessarily solve the strain series problem either. At low strain, recrystallization is not complete and deformation is partitioned between porphyroclasts and matrix, making any interpretation not easier than comparing strain gradients in single samples.
We update the notion in the text that, when referring to different strains. we actually consider the degree of deformation at the grain scale, either expressed by the gKam or aspect ratio - long axis alignment.

*reviewers comment: iii. <a> axis alignment. I really like the idea of using the intergrain <a> axis alignment (fig 8). I think this general approach could be very useful in other systems (ice in particular). I have a gut feeling that something useful can come from this analysis for this paper. However, at present the use of these data fall into the class of "fancy analysis that goes nowhere" in point 6 below. Part of the problem maybe that you show this only for the regime 3 sample. All the other key points in this paper come from comparison of the three samples and I think this is necessary to show the value of the <a> axis alignment. For example is the pattern shown in fig A2 the same in all three regimes? I think that a key point that may come from fig 8 is that the Y domains provide greater volumes (larger domains?) that can deform by <a> slip? The figure does not really show this well. I wondered whether it would be useful to show a map where domains connected by thresholds in <a>transparency are shown (and coloured as Y, B or other). This approach has some similarities to the grain boundary hierarchy approach (Trimby et al., 1998). Such data for the three regimes may*

*show key differences in size and connectivity of <a>transparent domains. These observations may then be linked to the interpretations about slip system activity.*

our reply: Considering the complexity of the manuscript , <a> axis alignment is left out and will be used elsewhere.

The purpose of the maps (former Fig.8, a a-axis alignment) was to show that Y-domains contain grains which, from the c-axis point of view are to be considered suitable for {m}<a>, however, still cannot deform homogeneously - and/or by {m}<a> alone because of the <a> misalignment across boundaries (which might be resolved by rigid body rotation as already suggested by Mancktelow, 1987). The purpose of the corresponding maps or low angle boundaries and misorientation axes was to show that in those grains low angle boundaries are present which are compatible with climb of edge dislocation of {m}<a>. Together, this indicates that, despite the y-domains seem to be high strain end-member, they most likely deform by by {m}<a> but also require gbs (rigid body rotation) - and this is something that helps understanding the girdle rotation with theta - also hinting at rigid body rotation during dislocation creep.

*reviewers commnt: iv. from the pdf*

*our reply: We are thankful for the detailed and helpful annotations. Most of those were adapted in the revised version of the manuscript, which we think improved the manuscript to a good part. One exception is the suggestion to omit the term "rotation" wrt the different postions of c-axes for different grain scale strains. As mentioned earlier, the "lack of strain series" has hopefully been clarified. We consider rotation not as an active term but as descriptive about the position of a direction.*

*Making the paper more accessible and understandable*
*reviewers comment: 1. Abandon "Texture". Throughout this paper the term "texture" is used with the meaning common in metallurgy and materials science. There is a very small community of geoscientists who use "texture" in this way. For the vast majority of the geoscience community "texture" means the spatial relationships of phases and microstructures. To most geoscientists texture is what you would see down a microscope (in a petrographic examination for example) and is broadly synonymous with the term microstructure. Textbooks in geoscience use the term texture in this way and there are specialised textbooks on textures that take this meaning (Metamorphic:, Barker, Shelley. Spry, Yardley & Mackenzie. Igneous Mackenzie & Mackenzie, McPhie et al., . Sedimentary: Holt, Scolle. Ore Minerals: Edwards, Taylor). I did a web of science search for "texture" in the title and "rock\*" in the topic. This recovers 888 papers. All but 4 of the newest 50, all but 2 of the oldest 50 and all but 3 of the highest-cited 50 use texture in the geoscience way rather than the metallurgical way (I've no time to look at 888). Having "texture" in the title is particularly problematic as most geologists/ petrologists will misinterpret it and most geophysicists won't know what it is. The terms "crystallographic preferred orientation" (CPO) or "lattice preferred orientation" (LPO) are much*

*better as they are explicit. If you want this paper (and the companion paper) to have wider readership, remove the word texture throughout and replace with CPO or LPO.*

our reply: We agree to use crystallographic preferred orientations in the title to omit any confusion for those readers who would interpret texture in the way it is commonly used by petrologists. However, we consider texture as a well known term to structural geologists and having lured the petrologists with new the title into reading our paper, we intend to keep the term texture in the main text. (LPO should in general not be used since L=lattice is a matter of definition.)

*reviewers comment: 2. A diagram to explain your reference frames. The names used for various orientations of c-axes on the pole figures are explained in words around lines 80-93. This is not very satisfactory: these need to be shown in a figure. A figure will act as a constant point of reference for readers looking at the results or discussion and trying to remember what particular grains or domains are. Remember also that most of Solid Earth readers will be unfamiliar with the general shear kinematics. Many will be unfamiliar with pole figures and especially with inverse pole figures. So a simple figure that has:*

1. *A cartoon of the general shear geometry and kinematics (like lower part of fig 1c in H&T, 2002)*

2. *A stereonet to show the various "domain" and "grain" orientations used in to describe the CPOs and the angle conventions (where domains relate to variable angles). Some key directions should be highlighted on a.*

3. *An IPF as used later in figs to show key crystal directions used.*

4. *This figure could also be used to relate names used in this paper to alternative names used in the literature – reducing text length and*

   *making these links clearer.*

*This figure will make the paper much easier to understand and will enable some of the figs that follow to be simplified.*

our reply:  We added a sketch to explain the experiment geometry, kinematic/strain reference frames as well as a polefigure displaying the texture domains. An IPF plot has been updated with annotations of popular crystal directions.

*reviewers comment: 3. Crystal axes. On lines 126-128 the indices and names of crystal axes are related. In the rest of the paper these two forms are interchanged. For example most of the figs use the indices and most of the text the names. This makes it difficult to read for those not familiar with indices in the trigonal system. I would stick to the names ([c], <a> etc). The words you have (l126- 128) are still needs but then stick to names in text and figs.*

our reply: We updated the figures and the text, with a few exceptions for non-trivial crystal directions. Some abbreviations however are only defined for planes e.g. {r} is equivalent to {10-11} but not the same as to <10-11> and we think that <r> provides an ambiguity. We try to stick with the abbreviations but unfortunately in some places the numerical form needs to be used.

reviewers comment: 4. *One point per grain vs one point per pixel. There are valid reasons to explore which of these choices are made. The complications added by showing a mixture of both makes this paper harder to follow. The choice makes no significant difference in any of the data presented (this is said somewhere but can't find it) so just pick one way. This will shorten the text and reduce the size/ number of figures. I would pick one point per pixel (all orientations).*

our reply: We settle to one orientation per pixel for bulk texture where we think this is the more useful information. In those cases where we can exclude any bias from a grain size dependency or where the calculation requires a definition of one single orientation for each grain, we use one orientation per grain. We agree that in most places differences between one orientation per grain and one orientation per pixel  are very subtle. Figures have been updated accordingly.

reviewers comment: 5 *IPF ref frame choice. A mixture of extended and folded IPF forms are used. I'm sure there is an argument as to which is used where, but this is another thing that will act as a significant point of confusion. On figs 2b and 8b I can see that it is important to have the folded form. In none of the other figures does it matter. The IPFs in fig 1b are mostly symmetric and little useful information is lost in folding them (also inconsistent to show these in extended form and not <a> pole figs on whole sphere). Actually I don't think there is any information in the IPFs in Fig 1 that you don't get from the pole figs. So why not just remove them. I would switch to folded IPF throughout and also make the colour schemes of IPF s the same (e.g figs 2a,b and 8b).*

our reply: With the only exception of (former) Figure 2b we prefer to display crystal directions  using the proper rotational group.  This has the advantage that a) the crystallography is honoured and the main reason is to be able to use a topologically correct colorcoding. In the case of Laue symmetry, we would either have the disadvantage to have color jumps or not being able e.g. to differentiate between {r} and {z} for example. As the reviewer noted, this becomes evident in Figure 2b which purposefully is updated and constructed for a higher pseudo symmetry since it has the purpose to display the the <a> direction without making a differentiate between <+a> and <-a>, and also when it comes to coloring crystal directions representing misorientation axes since it does not make any difference whether crystal A is rotated into B or the other way around and axes loose polarity. The colorcoding of the former Figure 2b was updated such that it will ignore also differences between {r/z} but that {m} will be represented in a coherent color without color jumps.

 Pole figures are upper hemisphere that implies that one would need to additionally plot <-a> to see all a-axes at the same time which would not add to our  observation that <a> aligns with the trend of the

plane of the highest shear stress. We agree that <+a> <-a> in the IPFs do not show any significant difference, however to stay consistent with the choice of the purely rotational point group, we prefer to show the entire 120° sector. The purpose of exploring the CPO by IPFs is that it becomes more evident that <a> shows the strongest alignment parallel to the direction of maximum shear stress (ISA-45) as well as the alignment of elastically soft directions parallel to the largest principal stress. While some of that could be guessed from the polefigures, we considered is more straightforward as well as systematic to read if off an IPF.

reviewers comments: 6 *Fancy analysis that goes nowhere. There is some very sophisticated analysis of the EBSD data in this paper. I'm sure it took a lot of time to work out and enact. Some of it really does not add anything (I cannot see any impact on the conclusions) and just makes the paper much harder to understand. In particular I see no value in figures 5, 7 and the two central columns of figure 9/A4 and the text that goes with these figs or figure elements. I have commented on fig 8 earlier. The key scientific points from Figs 9 and A4 are best made by the first and fourth columns alone. Indeed I made a composite of A4 to try and understand better the data myself. I've attached my simplified fig 9 to this review and I suggest a single figure of this form will help you explain the misorientation data and it's significance much more clearly. Similarly I made a compacted version of fig 6 (also attached) to understand better what these data show. The Solid Earth readership is not the place to explore the much more sophisticated analysis I have suggested you cut (see annotated pdf). It will take a lot of work to make these approaches intelligible and further work (and space) to make it clear what scientific value they add. If you can demonstrate the value of these approach I think you should publish them – but in a journal for a more specialist audience.*

our reply: We use "fancy" analyses mainly because we think that there are certain short comings with some of the standard types. The additional information in Figure 6 (former Figure 5) is  that one can easily relate a specific c-axis direction with an average grain property. Regarding the c-axis locations where most "rounded" and smallest grains are located, this representation adds to our interpretation.

Figure 8 (former Figure 7) presents a quantitative comparison of texture strength. Despite a few studies cited texture strength being related to the contribution of dislocation creep - and weakening to the action of gbs - quantitative approaches that cover a certain parameter space and are statistically sound are, to our knowledge lacking. The presented procedure enables both, exploring the parameter space and enabling a quantitative comparison of texture strength which can be directly related to the contribution of texture forming or weakening processes. Since we argue that dislocation creep - apart from causing a texture to strengthen - is also to blame for the development of a specific type of texture, we consider this analysis helpful.
The representation in polefigures always depends on the kernel density estimation which by itself is always a conservative approach, hence one should not make too many interpretations based on the "visual impression" of polefigures- especially since they are only a partial representation of the CPO.

So, essentially to demonstrate that e.g. texture strength varies according to given parameters, it is not sufficient to simply inspect a set of polefigures of a single direction. The presentation used here is to simplify the inspection of the results - which might be difficult to understand if presented in a table. The presented colorcoding scheme of misorientation axes has the great advantage of actually being able to unravel the ambiguities that arise from density contoured plots alone. This is best seen for the R-domains/grains where simply inspecting the density contoured misorientation axes in specimen and crystal coordinates would be misleading - however, the colorcoding can reveal that the high density of misorientation axes in specimen coordinates is actually a superposition of two distinct distributions, centered around [c]. We clarified this in the text and updated the figures to reduce the waste of space and redundant appreances.

The suggestion to introduce the misorientation color scheme concept in a separate publication has been considered but for this very purpose we do not think that this method is sufficiently "fancy" to merit an entire publication. We feel that there should be some place where this is actually first applied and referenced, such that it may help and stimulate other researchers who face similar problems.

We dropped pole figures for <a> and {r} in some evaluations, mainly - and as the reviewer noted - because the purpose of those was to show that there is not a significant change in those. While this is an important observation, we assume that the reader will trust our words when we note that there is no change for those directions.

reviwers commnt: 7. *Describe the "plots" at the point where they are used. The methods section includes the descriptions of the plotting approaches. When reading the results one has to constantly refer back to the methods. I would use the methods for the experiments and EBSD acquisition and processing (bring out of appendix) and describe the plots where they are presented in results. I think this will make the paper clearer and shorter.*

our reply: We agree and modified the text accordingly, rearranged the appendix, methods and result.

reviewers comment: 8 *Discussion: needs a figure and shortening. The discussion attempts to present a model to explain the CPO transitions the authors have documented. The effectiveness of the discussion will be improved by*

1. *Having a cartoon figure that reminds the reader of the key observations and the presents the model built open these (much like a graphic abstract).*

2. *Shortening (a lot) and tightening. The discussion is way too long. A paper can only make so many key points effectively and this discussion is far too expansive. The effect is the reader will take very little away from the paper. I have drafted what I think are key data points from my reading and suggest that these form the framework of discussion.*

our reply: We added a conceptual sketch of the concept of CPO development that we propose. We shortened the discussion as good as possible, however we feel some explanation need to remain for

those reader who are only partially familiar with some of the underlying concepts.

*reviewers comment: 9 Make real conclusions. The conclusions do not really give the reader any useful idea of what you have found out. They are a mess.*

our reply: We cleaned up the mess.

*reviewers comment: 10. Fold the appendices into the main text. I don't think you have anything that needs go into an appendix- it is all better in the main text – especially if number of figures reduced.*

our reply: We incorporated  the appendices into the main text and switched only the very technical part of the description of two methods (quantitative comparison of texture strength and pole figures of grain properties) to the appendix.

*reviewers comment: 11. Simple writing. The writing style is verbose. Sentences are too long. Complex phrases are used where simple ones would work better. I have highlighted examples on the annotated pdf. Overall the writing lacks clarity and needs considerable improvement.*

Structure and clarity of sentences has been improved ... hopefully.

*reviewers comment: 12. Literature interpretation versus observations. The discussion of the literature (particularly in the introduction and discussion) tends to focus on the interpretations of particular papers, sometimes without explicit reference to the data or observation that leads to the interpretation. For completeness and to enable better comparison to your new data literature citation needs to refer to both observations and interpretations.*

While cleaning up the ms, we corrected confusions arising from any insufficient distinction between reference and observation or interpretation.

Rüdiger Kilian, Renée Heilbronner, Basel, 21. of August 2017